Ontogenetic growth in the crania of Exaeretodon argentinus (Synapsida: Cynodontia) captures a dietary shift

Wynd Brenen bmwynd@vt.edu 1
Abdala Fernando 2 3
Nesbitt Sterling J. 1
1 Department of Geosciences, Virginia Tech , Blacksburg , VA , United States of America
2 CONICET-Fundación Miguel Lillo, Unidad Ejecutora Lillo , San Miguel de Tucumán , Tucumán , Argentina
3 Evolutionary Studies Institute, University of the Witwatersrand , Johannesburg , South Africa
Provete Diogo
Electronic publication date: 2022 Oct 21
Publication date: 2022
Volume: 10
Electronic Location ID: e14196
Received 2021 Dec 17; Accepted 2022 Sep 15
Copyright: ©2022 Wynd et al.
Copyright year: 2022
Copyright holder: Wynd et al.
License: This is an open access article distributed under the terms of the Creative Commons Attribution License, which permits unrestricted use, distribution, reproduction and adaptation in any medium and for any purpose provided that it is properly attributed. For attribution, the original author(s), title, publication source (PeerJ) and either DOI or URL of the article must be cited.
License URL: https://creativecommons.org/licenses/by/4.0/

Keywords: Cyndont, Exaeretodon, Traversodontidae, Ontogeny, Allometry, Crania, Ecological differentiation, Dietary ecology, Triassic, Ischigualasto formation

Funding: CONICET of Argentina National Research Foundation of South Africa Fernando Abdala was supported by the CONICET of Argentina and National Research Foundation of South Africa. The funders had no role in study design, data collection and analysis, decision to publish, or preparation of the manuscript.

==============================
Background

An ontogenetic niche shift in vertebrates is a common occurrence where ecology shifts with morphological changes throughout growth. How ecology shifts over a vertebrate’s lifetime is often reconstructed in extant species—by combining observational and skeletal data from growth series of the same species—because interactions between organisms and their environment can be observed directly. However, reconstructing shifts using extinct vertebrates is difficult and requires well-sampled growth series, specimens with relatively complete preservation, and easily observable skeletal traits associated with ecologies suspected to change throughout growth, such as diet.

Methods

To reconstruct ecological changes throughout the growth of a stem-mammal, we describe changes associated with dietary ecology in a growth series of crania of the large-bodied (∼2 m in length) and herbivorous form, Exaeretodon argentinus (Cynodontia: Traversodontidae) from the Late Triassic Ischigualasto Formation, San Juan, Argentina. Nearly all specimens were deformed by taphonomic processes, so we reconstructed allometric slope using a generalized linear mixed effects model with distortion as a random effect.

Results

Under a mixed effects model, we find that throughout growth, E. argentinus reduced the relative length of the palate, postcanine series, orbits, and basicranium, and expanded the relative length of the temporal region and the height of the zygomatic arch. The allometric relationship between the zygomatic arch and temporal region with the total length of the skull approximate the rate of growth for feeding musculature. Based on a higher allometric slope, the zygoma height is growing relatively faster than the length of the temporal region. The higher rate of change in the zygoma may suggest that smaller individuals had a crushing-dominated feeding style that transitioned into a chewing-dominated feeding style in larger individuals, suggesting a dietary shift from possible faunivory to a more plant-dominated diet. Dietary differentiation throughout development is further supported by an increase in sutural complexity and a shift in the orientation of microwear anisotropy between small and large individuals of E. argentinus. A developmental transition in the feeding ecology of E. argentinus is reflective of the reconstructed dietary transition across Gomphodontia, wherein the earliest-diverging species are inferred as omnivorous and the well-nested traversodontids are inferred as herbivorous, potentially suggesting that faunivory in immature individuals of the herbivorous Traversodontidae may be plesiomorphic for the clade.

Introduction

Ontogenetic growth characterizes multicellular life, with organisms shifting in absolute size and in the relative size of individual features (Huxley, 1932; Thompson, 1942; Gould, 1968; Gould, 1977; Gatsuk et al., 1980; Hochuli, 2001). Patterns of ontogeny have been repeatedly reconstructed in populations and species to estimate underlying constraints on development and evolution (Gould, 1968; Adams, 2000; Sanchez-Villagra, 2010; Goswami et al., 2012; Kolmann et al., 2018; Evans et al., 2019). A mathematical expression of shape change in ontogeny is frequently assessed based on allometries of numerous individuals from a growth series of the same species, which evaluates growth trends in correlations between different features (Gould, 1966; Cheverud, 1982; Alexander, 1985; Klingenberg, 1996; Voje et al., 2014; Kilmer & Rodríguez, 2017). Estimating allometries in conjunction with ecological observations allows for reconstructions of how organisms interact with their environments and how such interactions shift throughout growth, ultimately reconstructing the patterns and processes in the evolution of postnatal development.

Species-level allometry has been critical for studies of extinct species, as it is one of the few methods used to estimate patterns of development or even ontogenetic stage of individuals of an extinct species (see also paleohistology; Bailleul, O’Connor & Schweitzer, 2019), where developmental information is often lost to decay and taphonomic processes (e.g., Sampson, Ryan & Tanke, 1997; Huttenlocker & Abdala, 2015; Hoffman & Rowe, 2018; Griffin & Nesbitt, 2016; Griffin & Nesbitt, 2020; Griffin & Nesbitt, 2020; Hopkins, 2021). Studies that seek to reconstruct ontogeny in extinct species often use measurements that summarize size (e.g., skull length) in comparison with individual measurements (e.g., orbit length) to reconstruct allometric relationships for individual features, with the goals of understanding patterns of growth or to assess if multiple extinct species can be differentiated based on growth curves (e.g., Abdala & Giannini, 2000; Abdala & Giannini, 2002; Padian, Horner & De Ricqlès, 2004; Knoll, Padian & de Ricqlès, 2010).

Ecological shifts during ontogeny are common in vertebrate species, particularly dietary changes. Examples of ecological changes through ontogeny include shifts from insectivory (a form of faunivory) to herbivory in some extant lizards (Duffield & Bull, 1998), shifts from insectivory to carnivory in the American alligator (Dodson, 1975), increasing amounts of durophagy in the Nile monitor and the hyena (Lonnberg, 1903; Tanner et al., 2010), the shift from altriciality from parental care and nutrients to self-sufficiency in birds and mammals (Herring, 1985; Starck, 1993; To et al., 2021), or from planktivory to piscivory in some fishes (Ross, 1978). However, much of this body of work focuses on extant taxa and direct observations of their ecology and its changes, with fewer reconstructions of ecological change coupled with morphological change. Analyses of morphological change and associated ecological changes in extinct taxa reveal similarly few studies that document such shifts, but is far more difficult because of the incompleteness of the fossil record (Wang et al., 2017).

Previous studies of growth curves and morphological changes in extinct and extant species provide a theoretical backbone for reconstructing how morphological changes in size and shape may influence patterns of ecological change throughout growth. To estimate ecological change through growth in fossils, a feature that correlates to ecology must be targeted for study, and must have a relatively large sample with variation in size and/or shape. Furthermore, the fossils should have size-independent (diagnostic) characters to confidently identify a growth series as a single species instead of numerous taxa. Non-mammalian cynodonts—a paraphyletic group of primarily Triassic vertebrates—are an ideal group to evaluate ecological change through ontogeny because their record consists of well-preserved skulls with teeth that are often diagnostic to the species-level (Ruta et al., 2013). Analyses of non-mammalian cynodont ontogeny using allometry currently represent the earliest diverging epicynodonts, probainognathians, and one early diverging cynognathian (Parrington, 1936; van Heerden, 1972; Grine & Hahn, 1978; Grine, Hahn & Gow, 1978; Bradu & Grine, 1979; Abdala & Giannini, 2002; Jasinoski, Abdala & Fernandez, 2015; Jasinoski & Abdala, 2017a, with relatively few studies quantitatively reconstructing ontogenetic patterns for taxa that are well nested in the cynognathian subclades, especially traversodontids (Abdala & Giannini, 2000; Liu, 2007; Liu, Soares & Reichel, 2008). Cynognathia is well-suited to estimate correlations between ontogeny and ecology. This lineage includes early-diverging faunivorous members (e.g., Cynognathus crateronotus) and more well-nested herbivorous taxa (e.g., Exaeretodon argentinus). Furthermore, the well nested cynognathian clade Traversodontidae—a clade with faunivorous early diverging members and herbivorous later diverging members—reached relatively large body sizes and is known from tens of variably sized specimens; thus, this allows for a chronicling of change in growth and coordinated shifts in ecology, provided by multiple lines of evidence (Goswami et al., 2005; Abdala & Malabarba, 2007; Liu & Abdala, 2014; Kubo, Yamada & Kubo, 2017; Wynd et al., 2017; Hendrickx et al., 2020).

Here we describe the cranial ontogeny of the traversodontid Exaeretodon argentinus, a taxon known from many specimens from the Ischigualasto Formation of Argentina, and one of the largest bodied South American traversodontids (Filippini, Abdala & Cassini, 2022). Large differences in the size range of the Exaeretodon growth series (14.9–49.6 cm) may suggest that juveniles and adults experienced different trophic interactions, in the form of a dietary shift through growth (see Mittelbach, Osenberg & Leibold, 1988; Schiesari, Werner & Kling, 2009; Start, 2018). We describe the cranial ontogeny based on 16 measurements for 24 individuals, bolstered by previous allometry work (palatal measurements only) that posited Ischignathus as a junior subjective synonym of Exaeretodon (Liu, 2007). Of our sample, 11 specimens were collected from Argentina and brought to Harvard in 1965, and 13 are currently housed in Argentina. To account for the presence of deformation in a relatively low sample size, we implement a Generalized Linear Mixed Effects model (see Wynd, Uyeda & Nesbitt, 2021), to estimate allometric relationships used to infer ontogenetic change. With this reconstruction of ontogeny, we hypothesize a shift in diet from smaller to larger individuals.

Materials & Methods

Reviewing ontogenetic assessment in fossil synapsids

Estimating direct age in fossil synapsids requires numerous assumptions that are often not available or lack a necessary ground truth. One such method that has shown promise in numerous lineages is paleohistology (Kolb et al., 2015; Bailleul, O’Connor & Schweitzer, 2019), wherein lines of arrested growth are used to gauge age (i.e., skeletochronology), based on an assumption of annual cessation in growth due to resource availability, as shown in some extant lineages (e.g., Hutton, 1986). Osteohistology as a metric for skeletochronology can be complicated in lineages in which growth is rapid and continuous (e.g., Cynognathus crateronotus; Botha & Chinsamy, 2000). Although counting lines of arrested growth in histological samples can be used to assess age, much histological work assesses rates and patterns of growth and how histological patterns correlate to size (Botha & Chinsamy, 2000; Botha & Chinsamy, 2004; Botha & Chinsamy, 2005; Ray, Botha & Chinsamy, 2004; Botha-Brink, Soares & Martinelli, 2018). By combining histological data with size, relative ages can often be assumed, based on signals of decreased bone growth and the percent difference in size from the largest individual of the species (Botha & Chinsamy, 2005; Botha-Brink, Soares & Martinelli, 2018). However, using this method in the absence of additional growth data assumes that intrapopulation variation in size is relatively small, and that all relatively large bones would belong to developmentally older individuals. Studies that focus on cranial material alone are often unable to utilize osteohistology (but see Botha & Chinsamy, 2004), and are thus limited to using only size as a proxy for age. Fortunately, inferences regarding the appearance of osteological features (e.g., sutural morphology) and their correspondence with body size become possible when either closely related taxa are sampled, data include associated juveniles and adults (Jasinoski, Abdala & Fernandez, 2015; Jasinoski & Abdala, 2017a; Jasinoski & Abdala, 2017b; Hoffman & Rowe, 2018), or clustering methods are used on phylogenetically diverse datasets (O’Meara & Asher, 2016). In the absence of additional ontogenetic data, the common default is to evaluate the trajectory and patterns of growth via allometric reconstructions (e.g., Cheverud, 1982; Abdala & Giannini, 2000; Blob, 2006; Wynd, Uyeda & Nesbitt, 2021), rather than infer the developmental stage of a single specimen (e.g., juvenile vs adult). In the case of allometry, the statistical parameters that best describe the relationships between specimens (slope and intercept), represent a developmental trajectory, wherein differences between young and old individuals can be inferred, without ever attributing individual specimens to a particular developmental category.

There are currently no models that can estimate an age, exact or relative (e.g., juvenile vs adult), in specimens of Exaeretodon argentinus. However, under an allometric framework, we assume that basal skull length (BSL) is a proxy for age, and that in general, bigger skulls indicate older individuals. Assessing age based on size carries many assumptions with individual specimens, but with allometric models of shape change through growth, we assume that each individual slope and intercept for every feature (e.g., BSL vs MUL) allows for inferences to be made about growth. Throughout this manuscript, we avoid the use of relative age terms (e.g., juvenile, subadult, adult) as we cannot identify where in ontogeny these shifts occur, and we cannot confidently assign any region of our allometric models to relative ages. We do not know if our largest specimens reflect the maximum size of E. argentinus, and it is clear that we do not have samples that include the smallest neonates, as is the case in the tritylodontid, Kayentatherium wellesi (Hoffman & Rowe, 2018). The smallest individual of E. argentinus in our sample (BSL; 148.95 mm) is 30% the size of our largest individual (BSL; 495.88 mm); because of this, we assume that our specimens, and model interpretations, broadly sample the ontogenetic spectrum from young to old.

Specimens

All specimens studied herein were collected from the Ischigualasto Formation in San Juan Province, Argentina. Specimens housed at the MCZ were collected in the 1960’s by Alfred Sherwood Romer and his team. Specimens housed at the PVL and MACN were collected by Jose Bonaparte and other Argentinean paleontologists prior to 1994. Since their collection, a 1994 amendment to the Argentine constitution and the subsequent Archaeological and Paleontological Heritage Act of 2003 dictate that fossils belong to the provinces in which they were collected. As such, we recognize the traditional landowners of San Juan, and the Argentinean people to whom these fossils rightfully belong.

Data collection

All data herein were collected by the authors on physical specimens and photographs—due to COVID-19 limiting travel—of Exaeretodon argentinus (BMW: MACN, MCZ, PVL, PVSJ; FA: PVL), physical measurements were taken using digital calipers for skull length and features and a tailor’s measuring tape for skull length of specimens over 300 mm. For digital calipers, we recovered all measurements to the nearest hundredth of a millimeter, and for the tailors tape, we recovered measurements to the nearest tenth of a millimeter. For eight of the studied specimens (see Table 1), we took measurements from photographs using ImageJ v. 1.53c (Rasband, 1997). We used the ‘Set Measurement’ function on the scale bar of each photograph, to collect measurements to the nearest hundredth of a millimeter. For all analyses, measurements were rounded to the nearest tenth of a millimeter and log-transformed.

Table 1 Exaeretodon argentinus specimen condition.

Museum	Specimen #	Basal Skull length (mm)	Condition	In person/Photo?	
MACN	18125	443.6	Nasals broken dorsally, slight lateral shear	Photo	
MACN	18193	231.7	Unprepared, lateral and ventral views not represented in photographs	Photo	
MCZ VPRA	338-58M	210.6	Premaxilla missing	In person (BMW)	
MCZ VPRA	4468	320	Dorsal surface encased in plaster with lateral shear	In person (BMW)	
MCZ VPRA	4470	166.6	Premaxilla missing with lateral shear	In person (BMW)	
MCZ VPRA	4472	202.6	Dorsal surface encased in plaster	In person (BMW)	
MCZ VPRA	4478	156.1	Unprepared, right zygoma broken, left orbit broken, some anteroposterior crushing	In person (BMW)	
MCZ VPRA	4483	305	Premaxilla missing, lateral shear primarily in zygomas	In person (BMW)	
MCZ VPRA	4486	435	Braincase reconstructed with plaster	In person (BMW)	
MCZ VPRA	4493	225	Lateral shear with displaced muzzle bones. Lower jaws fused to skull	In person (BMW)	
MCZ VPRA	4494	265	Dorsal surface abraded	In person (BMW)	
MCZ VPRA	4505	312.3	Right zygoma missing	In person (BMW)	
MCZ VPRA	4781	149	Specimen used for thin-section, zygoma missing	In person (BMW)	
PVL	2056	178	Mediolaterally crushed with left side abraded	In person (FA)	
PVL	2067	294.3	Right side abraded with zygoma missing	Photo	
PVL	2080	334.2	Right side abraded with zygoma missing	Photo	
PVL	2082	265.7	Dorsoventrally crushed, dorsal braincase and part of the snout with plaster, left zygoma missing	In person (FA)	
PVL	2085	330.8	Left zygoma missing, unprepared	Photo	
PVL	2094	335	Portions of temporal area reconstructed with plaster	In person (FA)	
PVL	2109	274.3	Mediolateral crushing with posterior shear, right zygoma missing	Photo	
PVL	2473	328	Severe mediolateral crushing, left zygoma missing	In person (FA)	
PVL	2554	330	Dorsal skull eroded, basicranium and both zygoma missing	In person (FA)	
PVL	2565	230	Right anterior snout and zygoma missing	In person (FA)	
PVSJ	103	495.9	Slight lateral shear, ventral view not represented in photographs	Photo	

Following Abdala & Giannini (2000), we chose 15 measurements (see Fig. 1 and Table 2), as well as diastema length (DL)—following recent allometric studies of Thrinaxodon liorhinus (Jasinoski, Abdala & Fernandez, 2015)—and zygoma width, to incorporate the posterolateral zygomatic shelf to include additional masseteric attachment sites, for a total of 17 measurements (including BSL). We chose these measurements as they can be measured across various cynodonts (and other synapsids) without or with little ambiguity, and they summarize the overall changes in skull shape (Abdala & Giannini, 2000; Abdala & Giannini, 2002; Jasinoski, Abdala & Fernandez, 2015; Jasinoski & Abdala, 2017a). Furthermore, this set of measurements were all attainable through specimen photographs wherein only dorsal, lateral, and ventral photographs were taken.

Table 2 Measurement definitions.

Abbrv.	Feature	Measurement	
BB	Basicranial length	Length from the posterior extent of occipital condyles to the anterior extent of the pterygo-paraoccipital foramen	
BSL	Basal skull length	Tip of snout to the posterior extent of the occipital condyles	
BW	Maxillary bicanine widtha	Width of the snout, taken dorsally at the level of the canines	
DL	Diastema length	Length from distal edge of canine to mesial edge of the first postcanine	
IO	Interorbital distancea	Minimum distance between the orbits	
MUL	Muzzle length	Tip of the snout to the anterior extent of the orbit	
OD	Orbit diameter	Diameter of the orbit	
OL	Orbit length	Anteroposterior length of the orbit	
OW	Occipital plate widtha	Maximum width between the opisthotics	
PAL	Palate length	Tip of the snout to the posterior extent of the secondary palate	
PD	Posterior postcanine distancea	Maximum width between the lingual margins of the distalmost postcanines	
SW	Skull widtha	Maximum width of the skull	
TEL	Temporal region length	Posterior extent of lambdoidal crest to the anteriormost point of the temporal fenestra	
TP	Transverse process widtha	Maximum width between the lateral extent of the transverse processes.	
UP	Upper postcanine length	Length of the postcanine series. Mesial and distal margins of alveoli are appropriate with missing teeth	
ZH	Zygoma height	Maximum height of the zygoma, including both squamosal and jugal	
ZW	Zygoma width	Lateralmost margin of squamosal to medial margin of the anteriormost tip of squamosal	
Notes.

a When elements are missing on one side, it is presented as the duplication of the measurement on one side of the skull.

Taphonomic distortion is a critical issue in reconstructing patterns of allometry in extinct animals that often results in measurement omission or estimation (Brown, Arbour & Jackson, 2012; Brown & Vavrek, 2015; Wynd, Uyeda & Nesbitt, 2021). Fortunately, skull length in E. argentinus is typically preserved and not prone to as much distortion, partly due to regions where many bones contact one another with a relatively large surface area to form resistant structures (e.g., snout vs zygoma). On the other hand, some areas of the skull, particularly regions surrounding the zygomatic arches, where bones are thin and are not supported on all sides by other structures, are more prone to taphonomic deformation.

Figure 1 Reconstruction of the skull of Exaeretodon argentinus indicating the measurements taken for this study.

Reconstructions are based on MCZ VPRA-4483. The skull is shown in (A) dorsal, (B) ventral, and (C) right lateral views. Abbreviations: BB, basicranial length; BSL, basal skull length; BW, maxillary bicanine width; DL, diastema length; IO, interorbital distance; MUL, muzzle length; OD, orbit diameter; OL, orbit length; OW, occipital plate width; PAL, palate length; PD, posterior postcanine distance; SW, skull width; TEL, temporal region length; TP, transverse process width; UP, upper postcanine tooth row length; ZH, zygoma height; ZW, maximum zygoma width.

For each of the 16 measurements of the skull (see Fig. 1 and Table 2), we scored them as undistorted (0) or distorted (1). Assessing a measure as ‘distorted’ was largely based on deviations from bilateral symmetry (e.g., MCZ VRPA-4469 and PVL 2473), identifying elements that were either misplaced (e.g., misaligned sutures as in MCZ VPRA-4470) and/or there was evidence of twisting or shearing, which was largely in the temporal region in Exaeretodon argentinus (e.g., MCZ VPRA-4470, 4483, and 4493). We did not collect measurements for features that were entirely missing; for example, both zygomatic arches are missing on MCZ VPRA-4781, and as such, skull width (SW) was not measurable. However, when only one side is missing (e.g., MCZ VPRA-4505), we estimate skull width (SW), as well as other features, based on bilateral symmetry as twice the width from the lateralmost margin of the zygomatic arch to the midpoint of the sagittal crest. When only a portion of a feature is missing (e.g., missing premaxilla when measuring muzzle length; MUL), the measurement is taken as is and scored as distorted. In the case of a missing premaxilla (MCZ VPRA-338-58M), we measure basal skull length (BSL), based on the anteriormost extent of the maxilla when the canine or canine alveolus is present, as the anterior extent of the premaxilla is roughly coincident with the anterior extent of an intact maxilla. For features that are distorted, we take the measurements as is and score those measurements as distorted (1) in our dataset.

To test if our variables follow normal distributions, we ran a Shapiro–Wilk test on each of our measurements. Only diastema length (DL) was found to have a non-normal distribution (see R supplement), which is likely a result from 11 of the 14 measurements being considered distorted, and so there was likely some degree of non-random preservation in the distorted samples. All of our other measurements returned a p-value >0.05, indicating that their distributions are not significantly different from a normal distribution. Because of this, we interpret that our sample largely represents a growth series, with individual-level variation across the population. Additionally, we assume that there is no sexual size dimorphism in our sample because our measurements do not follow individual bimodal distributions. Additionally, we did not observe any discrete characters that exhibit patterns of variation consistent with sexual dimorphism for Exaeretodon argentinus. However, it is still possible that sexual dimorphism is present and influences our ontogenetic hypotheses, but without clear justification for potentially dimorphic characters, we refrain from any sex-based interpretations of ontogeny.

Allometric reconstruction

We reconstruct individual allometries for each of the 16 measurements against basal skull length (BSL; Fig. 1). Ordinary least squares (OLS) regression is commonly used for undistorted measurements whereas generalized linear mixed models (GLMM) have been explored on datasets that include undistorted and distorted measurements (Wynd, Uyeda & Nesbitt, 2021). The existing sample of E. argentinus specimens often have distorted features that would lead to dropping those specimens from analysis, which leads to a mean number of available specimens of 16, rounded. On average, dropping specimens would result in the omission of at least eight specimens per feature, or the estimation of the feature dimensions which introduces investigator biases into the data, if such specimens are ultimately included. We use a GLMM, which can sufficiently account for additional variation due to distortion (Wynd, Uyeda & Nesbitt, 2021). We followed previous methodology and treated the feature of interest as a fixed effect and distortion as a random effect (Wynd, Uyeda & Nesbitt, 2021). This model assumes random variation due to distortion in the sample and fails to adequately estimate the y-intercept when the distortion is non-random and produces a similar condition in all distorted specimens; however, this caveat does not affect the reconstructed slope (Wynd, Uyeda & Nesbitt, 2021). Variation due to distortion is not uniform in E. argentinus (see Fig. 2), and thus the GLMM is an appropriate model that allocates variation due to deformation, when present (see Wynd, Uyeda & Nesbitt, 2021 for more thorough discussion of model sensitivity). We indicate a model as having positive or negative allometry when the recovered slope is significantly different from 1.0. Values that cannot be statistically differentiated from a slope of 1.0 suggest that a feature is isometric and does not change its relative size, in relation to the length of the skull, throughout the lifetime of the organism. We use a p-value <0.05 as our cutoff for statistical significance, but in cases where the p-value is marginally significant (herein estimated as a p-value ≲ 0.10, but >0.05), we report that feature as either reflecting isometry or positive/negative allometry, assuming that the feature would likely be significantly different from isometry with greater sample size. We performed both a linear regression on only undistorted specimens, and a generalized linear mixed model on the dataset including distorted and undistorted specimens, as an additional method to evaluate model congruence and how model selection (OLS vs GLMM) affects inferences about growth. All analyses were performed in the R statistical environment v. 4.0.2 (Team RC, 2013) and the generalized linear mixed models were performed using the lme4 package v. 1.1.23 (Bates et al., 2007). We report the returned parameters for each of the models (see Table 3) to evaluate the overlap between the two models, linear regression and generalized linear mixed model. The lme4 package will return a singular fit warning if one of the effects (fixed or random) has a variance near 0.0—the model is able to identify variation that exists in the distorted sample but not in the undistorted sample or vice versa—indicating that either one of the groups (distorted or undistorted) lacks necessary sample sizes to estimate the variance in the random effect, or that the residuals of the distorted specimens fall within—or very near—the residuals of the undistorted sample. As our random effect is meant to account for taphonomic variation and remove that variation from estimations for the slope, a singular fit is not necessarily a negative result for our model but does require more investigation as to the reason for the lack of variance in the random effect. To evaluate whether there is an impact of body size on the presence of taphonomic deformation, we compare the length of the skull to the total number of distorted features on a single individual, though we only include specimens that can be examined on all sides (Fig. 3).

Figure 2 Skulls of Exaeretodon argentinus displaying varying degrees and forms of deformation.

(A) MCZ VPRA-4470; (B) MCZ VPRA-4468; (C) MCZ-VPRA-4486; (D) PVL 2473; (E) PVSJ 103. Skulls are shown in dorsal (A, C, D), ventral (B), and right lateral (E) views. Scale bar equals 10 cm.

Morphological comparisons

We include brief morphological comparisons between semaphoronts—individual specimens attributed to different growth stages of an ontogeny—to evaluate ontogenetic shifts not readily recognized by linear allometries. Such comparisons primarily focus on sutural morphology and the general morphology of muscle attachment sites.

Dental microwear

To further test whether smaller individuals of Exaeretodon argentinus were ecologically similar to larger specimens described in Kubo, Yamada & Kubo (2017), we include a qualitative discussion of microwear from the labial surface of an isolated upper left postcanine tooth. This specimen is cataloged as MCZ VPRA-4470 (BSL∼16.6 cm; ∼33% maximum BSL) and is consistent in size with the postcanine alveoli of the skull of MCZ VPRA-4470 but does not presently fit into any of the open alveoli, which are anteroposteriorly constricted. The tooth of MCZ VPRA-4470 is the only isolated tooth available for this sample, and was well-preserved labially, but the occlusal surface was obstructed by matrix. Because of this, we molded the labial side of the isolated tooth, using methods outlined by Bestwick et al. (2020); e.g., molds were created with President Jet Regular Body polyvinylsiloxane (Coltène/Whaledent Ltd., Burgess Hill, West Sussex UK) and the first mould taken from the mesial and distal regions of the labial surface of the specimen were discarded to remove any surface material (e.g., dirt, glue parts) and our casts were made on the second molds. Two casts (Smooth-On Smooth-Cast 300 liquid plastic) were made from each mold, for a total of four individual casts of the labial side of the isolated tooth to ensure that the patterns evidenced under the SEM are reflective of the tooth morphology and are not an artifact of the molding and casting process. We used a Hitachi TM3000 tabletop thermionic (tungsten filament source) SEM with polepiece backscattered electron solid-state detector housed in the Virginia Tech Department of Geosciences. Tooth casts were affixed to a stable mount with a combination of clay and double coated carbon conductive tape, to ensure specimen stability and increased conductivity.

Table 3 Summary statistics for Generalized linear mixed effects model (GLMM) and ordinary least squares regression (OLS).

Feature	n GLMM	D GLMM	Singular fit?	α GLMM	β GLMM	P (α = 1)	n OLS	α OLS	β OLS	P (α = 1)	R 2	Trend	
MUL	22	14	false	0.96405	−0.27381	0.5516	8	0.9364	−0.1934	0.55	0.9243	Isometric	
PAL	12	9	true	0.843915	0.005033	0.0792	3	0.9889	−0.329	0.937	0.9697	Isometric/ Negative	
OL	19	10	false	0.7321	−0.1688	0.0248	9	0.564	0.3006	0.0234	0.6203	Negative	
IO	19	14	false	0.9488	−0.4668	0.73	5	0.6144	0.3495	0.329	0.4389	Isometric	
TEL	20	12	true	1.1761	−0.8304	0.0421	8	1.2077	−0.8982	0.1487	0.9271	Positive	
SW	21	12	false	1.13435	−0.39337	0.16	9	1.1553	−0.4326	0.379	0.8572	Isometric	
BW	22	11	false	0.9877	−0.4419	0.916	11	1.1296	−0.8099	0.4736	0.8008	Isometric	
UP	16	4	true	0.5286	0.6217	0.000685	12	0.5026	0.6808	0.0025	0.5845	Negative	
PD	14	8	true	0.9652	−0.6719	0.836	5	1.4392	−1.8318	0.414	0.6902	Isometric	
TP	12	10	false	0.83507	−0.12841	0.101	2	0.82948	−0.05587	0.175	0.9967	Isometric/ Negative	
OD	9	7	true	0.195	1.102	0.119	2	−0.8149	3.5133	0.575	0.1112	Isometric	
OW	14	6	false	1.1273	−0.6721	0.3359	6	1.1978	−0.8085	0.461	0.824	Isometric	
BB	12	5	true	0.5238	0.4759	0.00606	7	0.6402	0.2059	0.0966	0.6715	Negative	
ZH	15	12	false	1.3084	−1.3293	0.08026	3	1.5675	−2.0717	0.0993	0.9704	Isometric/ Positive	
DL	14	11	false	1.1151	−1.2797	0.806	3	0.9384	−0.7213	0.905	0.678	Isometric	
ZW	12	3	True	1.1894	−1.4933	0.623	9	1.059	0.4493	0.899	0.4098	Isometric	
Notes.

GLMM used as the model to infer the allometric trends.

n the number of specimens used for the GLMM and OLS

D the number of distorted specimens coded for the GLMM

α slope of the regression line for both GLMM and OLS

β y-intercept for both GLMM and OLS

P the p-value for if the regression line is significantly different from a line with a slope of 1

R2 the proportion of the data explained by the regression line under the OLS

Bolded entries reflect statistically significant deviations from isometry.

Figure 3 Plot comparing the skull length and number of deformed features.

Used as a proxy to compare degree of deformation between specimens.

Taxonomic identification

Exaeretodon argentinus (Cabrera, 1943)	

Holotype MLP 43-VII-14-2, incomplete left mandibular ramus.

Referred Diagnosis We follow previous diagnoses to identify each of our specimens as Exaeretodon argentinus. Exaeretodon is differentiated from all other traversodontids, except Siriusgnathus niemeyerorum, by “[v]ery large traversodontids lacking an internarial bar; upper postcanines with a well-developed posterolabial accessory cusp and extensive shouldering resulting in a separation between a labial lobe and a lingual one (including the occlusal basin); …divergent zygomatic arches; well-developed descending process of the jugal…” (Liu & Abdala, 2014, pg. 269). Exaeretodon can be differentiated from Siriusgnathus by a more distally placed labial accessory cusp of the upper postcanine; rostrum length equal or subequal to the temporal region; a more posteriorly positioned postorbital bar; anteriormost portion of the squamosal reaching the level of the postorbital bar; contact of the squamosal and jugal forming a depression; lambdoidal crest forming a concavity; and a comparatively anteroposteriorly long basicranium (Pavanatto et al., 2018). Exaeretodon argentinus can be further differentiated from E. riograndensis based on the absence of prootic crests, and more varied numbers of postcanines in ontogeny (Abdala, Barberena & Dornelles, 2002).

Referred specimens MACN: 18125, 18193. MCZ VPRA: 338-58M, 4468, 4470, 4472, 4478, 4483, 4486, 4493, 4494, 4505, 4781. PVL: 2056, 2067, 2080, 2082, 2085, 2094, 2109, 2473, 2554, 2565. PVSJ 103.

Comments The holotype of Exaeretodon argentinus was originally described as Belesodon? argentinus based on an incomplete left ramus of the dentary (Cabrera, 1943). In the same volume, but following the description of Belesodon? argentinus, the genus Exaeretodon was erected with the type species being E. frenguelli. However, later work has suggested that Belesodon? argentinus is actually a specimen of Exaeretodon frenguelli, and as such, based on the principle of priority, E. frenguelli has been regarded a nomen dubium of E. argentinus (sensu Liu, 2007). The type specimen refers only to an incomplete ramus of the dentary and is largely uninformative to E. argentinus specimens that either lack mandibles or have mandibles still bound to the crania by matrix. We extensively use MACN 18125, MCZ VPRA-338-58M, and MCZ VPRA-4483 as reference specimens, as they consist of complete or nearly complete crania with some or all postcanine teeth in place; these specimens lack associated mandibles but were compared alongside other MCZ specimens that retained mandibles. Exaeretodon crania tend to have no associated mandible, or the mandible is preserved in contact, obstructing views of the palate and dentition, meaning that our primary specimens for comparison lacked mandibles. Furthermore, although they range in size (from 210.57 mm to 443.6 mm), they retain diagnostic dental characters, strongly suggesting they are of the same species. All of our referred specimens, whose diagnostic features are consistent with MACN 18125, MCZ VPRA-338-58M, and MCZ VPRA-4483 and lack prootic crests, are attributed to E. argentinus.

Results

Allometry

We recover necessary minimum sample sizes of distorted specimens for the GLMM for all measurements, except for length of the upper postcanine series (UP), and maximum width of the zygomatic arch (ZW; see Table 3). For the undistorted measurements, we lack minimum sample sizes for length of the palate, maximum width of the transverse processes of the pterygoids, diameter of the orbit, maximum height of the zygomatic arch, and diastema length. Based on the lack of minimum sample sizes for undistorted specimens, our sample of Exaeretodon argentinus crania tend to possess taphonomic deformation. We find no significant relationship between basal skull length and taphonomic distortion (Fig. 3; α = −0.85, P (α = 0) = 0.84, R2 = 0.0023). Our GLMM is able to estimate allometric relationships with narrower confidence intervals, compared to the OLS, especially when there are more distorted than undistorted measurements (see Fig. 4). For all features we find that the confidence intervals of the GLMM overlap the regression line (See Figs. S1–S13). Orbit length (OL), upper postcanine length (UP), and basicranial length (BB) were all found to reflect negative allometric relationships under both the OLS and GLMM (Table 3; OL, α = 0.73, P(α = 1) = 0.025; UP, α = 0.53, P(α = 1) = 0.00069; BB, α = 0.52, P(α = 1) = 0.0061). Palate length (PAL) and width of the transverse processes (TP) showed an isometric relationship under OLS, but a negative allometric relationship under GLMM, based on marginally significant p-values (Table 3; PAL, α = 0.84, P(α = 1) = 0.079; TP, α = 0.84, P(α = 1) = 0.10). Only temporal length (TEL) showed a positive allometric relationship under both models, and zygoma height (ZH) was positive only under GLMM with a marginally significant p-value (Table 3; TEL, α = 1.18, P(α = 1) = 0.042, ZH, α = 1.31, P(α = 1) = 0.080). All other measurements returned coefficients of allometry that were not significantly different from a slope of 1 and interpreted as isometric. However, those with marginally significant p-values are herein considered, due to the relatively small sample size. Proportional differences between small and large individuals capture more nuanced shape differences in the growth of Exaeretodon argentinus (Fig. 5).

Figure 4 Representative plots comparing the fit of the generalized linear mixed effects model (GLMM), to the ordinary least squares regression.

95% confidence intervals are reported as dotted lines in corresponding colors. (A) Similar fit with overlapping confidence intervals. GLMM has confidence intervals that are more narrow than those for the ordinary regression. (B) A similarly positive slope for both models with a poorly fit ordinary regression. (C) Contrasting slopes between models with an overall poor fit for both models, but the GLMM is able to incorporate specimens that indicate that the slope is not negative.

Figure 5 Reconstruction of ontogenetic change in the skull of Exaeretodon argentinus.

Upper (A, B) and left (C) reconstruction is based on MCZ VPRA-4470, and lower (A, B) and right (C) reconstruction is based on MCZ VPRA-4483. Reconstructions are shown in (A) dorsal, (B) ventral, and (C) lateral views. Reconstructions are not reflective of actual size differences between individuals but are meant to indicate the major morphological differences between large and small individuals. Black silhouettes reflect the minimum skull size for E. argentinus (∼15 cm), compared to the largest size (colored reconstructions). Bones are colored to reflect homology.

Morphology

The allometric relationships discussed herein offer an insight to how general skull morphology covaries with body size in Exaeretodon argentinus (see Fig. 5). A particular challenge in using only allometric relationships between linear measurements is an assumption that all ontogenetic shifts over a growth series are related to size of a feature, and do not regard the general morphology of the feature (e.g., muscle scars or suture morphology). Herein, we provide a brief description of the observed morphological differences that would not be captured by allometric relationships of linear measurements between small and large individuals of E. argentinus.

Snout The snout of Exaeretodon argentinus is a complex of tightly sutured bones including the premaxilla, maxilla, nasal, prefrontal, lacrimal and a small contribution from the jugal. The snout often shows minimal deformation, usually in the form of slight twisting or shearing (e.g., MCZ VPRA-4470, 4483, and 4781). The snout length (MUL) of E. argentinus shows an isometric relationship with basal skull length (BSL) throughout ontogeny. When viewing morphological change holistically over the growth series, the snout often appears to be relatively shorter in larger individuals, which is likely due to positive allometry of the temporal region (TEL), with a negative allometry of the orbits balancing the anteroposterior growth of the temporal region (Fig. 5). Among our sample, several specimens either have some degree of matrix still on the bones or regions of the crania reconstructed with plaster (e.g., MCZ VPRA-4486); these conditions often made it difficult to accurately interpret the bounds of sutural contacts. Therefore, we only report cases where we can confidently interpret sutural morphology (Fig. 6), and we do not report contacts where breaks or matrix make portions of sutural morphology unrecognizable (e.g., Fig. 6C).

Figure 6 Sutural morphology in Exaeretodon argentinus.

(A) MCZ VPRA-4470 (BSL: 166.6 mm) dorsal view of the interorbital region and posterior snout, with a simple contact between the nasals-frontals-prefrontals. (B) MCZ VPRA-338-58M (BSL: 210.6 mm) dorsal view of the interorbital region and posterior snout, showing simple sutural contacts between the nasals-frontals-prefrontals. (C) MCZ VPRA-4483 (BSL: 305 mm) dorsal view of the interorbital region and posterior snout, with most sutural contacts occluded by matrix/breakages, but the nasofrontal suture is clearly interdigitated. (D) MCZ VPRA-4483 in dorsal view (close up of C) with interdigitated suture between the nasal and lacrimal anterolateral to the right orbit. (E) MCZ VPRA-4483 lateral view of the orbit and anterior portion of the zygoma showing interdigitation between the postorbital-jugal contact. (F) MCZ VPRA-338-58M left zygoma in lateral view with simple contacts between the postorbital-jugal and jugal-squamosal. Scale bar equals two cm in all panels. Abbreviations: Fr, frontal; Ju, Jugal; La, lacrimal; Na, Nasal; Po, postorbital; PrF, prefrontal; Sq, squamosal.

Within the snout, sutural morphology appears to change throughout ontogeny, with larger individuals bearing superficial interdigitation wherein the sutures primarily zig zag to form thin fingers that interlock with one another (Figs. 6C–6D). Sutural complexity in larger individuals varies based on orientation, wherein sutures oriented transversely to the skull tend to bear largely sinusoidal morphology with few examples of sutural fingers folding in on one another to form interlocking elements (Fig. 6D). Conversely, smaller individuals tend to have simpler suture morphology, superficially, where bones appear to abut one another without any complex interlocking, clearly represented by the simple sutural contacts between the nasals and surrounding elements (Figs. 6A–6B). The contacts between the nasals and the frontals, prefrontals, jugals, and lacrimals then shift towards expanded interdigitation in larger individuals, particularly in the regions in which three bones contact one another. The suture between the nasals and the maxillae appears simple across body size; nevertheless, maxillae are only displaced from the nasals in small crania (e.g., MCZ VPRA-4470 and MCZ VPRA-338-58M), which may indicate a more complex and tight sutural contact with internal interdigitation in larger individuals. In smaller individuals, the nasals show an inverse-U shape posteriorly, such that the lateral margins of the nasals extend more posteriorly, resulting in an anterior extension of the frontals fitting between the nasals (Fig. 6B). This relationship appears to change through ontogeny, wherein the nasofrontal contact transitions from a U-shape in smaller individuals to a coronal contact, where the nasals do not expand posteriorly beyond the anterior extent of the frontals.

Palate The palate consists of parts of the premaxilla, maxilla, palatine, and anterior contributions of the pterygoid. The vomer in E. argentinus is hidden in ventral view by the secondary palate. The palatal region shows an overall reduction in the anteroposterior length of the upper postcanine series, as well as an extension of the length of the diastema between the canine and postcanines, consistent with the argument that E. argentinus lost postcanine teeth anteriorly more quickly than they were erupted posteriorly, though at a slower rate than in E. riograndensis (Abdala, Barberena & Dornelles, 2002). Although there is discrepancy between the linear regression and the GLMM, the overall length of the palate shows a negative allometry, under the GLMM (see Table 4). As with the snout, the palate is often resistant to deformation, except for some lateral shearing in some specimens (e.g., MCZ VPRA-338-58M, 4468, and 4470). Sutural morphology in the palate appears to remain superficially similar throughout ontogeny. However, few large specimens, with the exception of MCZ VPRA-4483, have fully prepared palates to evaluate ontogenetic changes with confidence. Nevertheless, where visible, the maxilla-palatine, maxilla-premaxilla, and palatine-pterygoid contacts remain simple with no evidence of interdigitation externally.

Table 4 Allometric comparisons between cynodonts.

Feature	Galesaurus planiceps	Thrinaxodon liorhinus	Diademodon tetragonus	Massetognathus pascuali	Exaeretodon argentinus	Chinquodon theotonicus	
MUL	Iso (0.94)	Pos (1.17)+	Iso (1.01)	Neg (0.94)−	Iso (0.96)	Iso (1.02)	
PAL	Iso (0.96)	Pos (1.15)+	Iso (0.98)	Neg (0.83)−	Nega (0.84)−	Pos (1.12)+	
OL	Neg (0.63)−	Neg (0.65)−		Neg (0.91)−	Neg (0.73)−	Iso (0.97)	
IO	Iso (1.09)	Iso (0.99)	Iso (0.92)	Iso (1.09)	Iso (0.95)	Iso (1.09)	
TEL	Pos (1.96)+	Pos (1.41)+		Pos (1.25) +	Pos (1.18)+	Pos (1.12)+	
SW	Pos (1.56)+	Iso (0.95)	Posa (1.17)+	Pos (1.30)+	Iso (1.13)	Pos (1.12)+	
BW	Iso (1.09)	Iso (1.1)	Iso (1.02)	Iso (0.99)	Iso (0.99)	Pos (1.24)+	
UP	Iso (0.9)	Iso (0.93)	Neg (0.93)−	Neg (0.83)−	Neg (0.53)−	Iso (0.93)	
OD	Iso (1.02)	Neg (0.87)−	Neg (0.69)−	Neg (0.73)−	Iso (1.1)		
OW	Iso (0.99)	Iso (0.9)	Pos (1.28)+	Iso (1.12)	Iso (1.13)	Iso (1.11)	
BB	Iso (0.73)	Iso (0.92)		Neg (0.87)−	Neg (0.52)−	Pos (1.28)+	
ZH	Pos (1.83)+	Iso (1.17)	Pos (1.21)+	Pos (1.37)+	Posa (1.31)+	Pos (1.24)+	
Notes.

Coefficients for E. argentinus are based on the GLMM, whereas coefficients for the other taxa are based on reduced major axis regression (data taken and adapted from Jasinoski & Abdala (2017a).

a Marginally significant (0.05 <p ≲ 0.1),

Feature names are listed in Table 2.

Abbreviations Iso Isometry

Neg Negative allometry

Pos Positive allometry

+ Positive allometries

− Negative allometries

Temporal region The temporal region consists of the posterior extent of the frontals, the postorbitals, parietals, jugals, squamosals, and quadrates. We exclude contributing elements of the braincase, as the temporal region is interpreted as the temporal fenestra and the surrounding elements, which would have served as attachment sites for the m. masseter, temporalis, and other feeding related musculature (m. pseudotemporalis and pterygoideus; see Lautenschlager et al., 2017). The temporal region is the most susceptible to deformation across specimens of E. argentinus, including lateral shearing, displacement of elements, or even an entire zygomatic arch being lost to any combination of breakage or abrasion (e.g., MCZ VPRA-4470, 4486, and 4505). The length of the temporal region and maximum height of the zygoma both show an overall positive allometric pattern (TEL, p = 0.042, significant; ZH, p = 0.080, marginally significant, H0: α=1), whereas the overall width of the skull with respect to its total length is isometric. Notably, the rate of expansion (=coefficient of allometry) of the zygoma height (α = 1.31) exceeds that of the length of the temporal region (α = 1.18). Cranial musculature of both extinct and extant synapsids is reconstructed with the temporal fenestra being almost entirely occupied by the m. temporalis (Gregory & Adams, 1915; Barghusen, 1973; Hopson, 1994; Sidor, 2001; Kemp, 2005; Lautenschlager et al., 2017). This indicates that the sites for muscle attachment and occupation were both increasing throughout the lifetime of an individual, but that the expansion of the zygomas was outpacing the length of the temporal fenestra (Table 3 and Fig. 7). Notably, the 95% confidence intervals for temporal length are relatively narrow and do not overlap the mean for zygoma height, but the confidence intervals for zygoma height are wider and do overlap the mean of temporal length. This indicates that temporal length is growing significantly differently from zygoma height, but zygoma height is not growing significantly differently from temporal length; although greater sample sizes will likely result in narrower confidence intervals that further separate temporal length from zygoma height in Exaeretodon argentinus, ultimately resulting in non-overlapping confidence intervals. Additionally, larger individuals (e.g., MCZ VPRA-4505 and MCZ VPRA-4483) possess a lateral expansion of the zygomatic arch, at its most dorsal and posterior extent, that forms a lateral overhang that increases the total surface-area of the zygomatic arch. However, zygoma width was recovered as isometric, likely due to the anterior extent of the squamosal, which is depressed medially. The squamosal is deflected more medially than the more laterally projected jugal, which results in a mediolaterally expanded zygoma in smaller specimens that lack a lateral shelf of the squamosal. By modeling the width of the entire zygoma and not just the width of the jugal or squamosal, the expanded width for small individuals ultimately forces a lower slope and thus an isometric relationship with skull size.

Figure 7 Regression lines between skull width, temporal region length, and zygoma height with basal skull length, based on results of the GLMM.

To show relative rates of growth, black circles indicate points in which zygoma height becomes larger than skull width and temporal region length (both of which occur at skull sizes over 1 m in length, which is not known for E. argentinus). Skull width here represents isometry. The black semicircle indicates that the zygomas are smaller than the temporal region and skull width in juvenile individuals, based on the lower y-intercept (βGLMM).

Nearly all contacts between bones of the temporal region show relatively simple straight suture patterns in smaller individuals, that transition to interdigitating sutures in larger individuals (Figs. 6E–6F). The primary exception exists along the squamosal-jugal contact, which remain simple in both small and large individuals.

Braincase The braincase of E. argentinus consists of the basioccipital, opisthotic, basisphenoid, and prootic. The braincase is often well-preserved with lateral shear being most of the deformation observed (e.g., MCZ VPRA-4493). Like the snout, the braincase is a region of many tightly sutured bones that produce a relatively resistant structure. The anteroposterior length of the braincase (BB) shows a negative allometric trend, which is generally consistent with the notion that regions of the skull that house the brain and sensory organs often show negative allometric relationships in vertebrates (Howland, Merola & Basarab, 2004). In contrast, the occipital plate width (OW) has an overall isometric pattern, which is perhaps related to the fact that the measurement (or variable) is reflecting the braincase width but also the enlargement of the splanchnocranial apparatus. Isometry in the width of the skull would indicate that the braincase is growing at a constant rate mediolaterally and potentially covarying with other cranial features linked to jaw placement (e.g., width of transverse processes; TP or possibly even zygoma width as mentioned above). However, the anteroposterior length of the braincase (BB) shows a negative relationship, indicating a relative reduction in the overall size of the braincase through growth. Many larger specimens of E. argentinus have ventrally underprepared braincases making accurate reconstructions of suture morphology difficult to assess. What is available suggests overall similar sutural patterns as elsewhere in the skull, wherein braincases of smaller individuals (e.g., MCZ VPRA-4781) bear simple sutures with little to no interdigitation both ventrally and on the lateral wall. Few ventral sutures are evident in larger skulls, with the exception of the opisthotic-squamosal contact, which is consistently simple and lacks any evidence of interdigitation. In contrast, the parietal-squamosal contact bears interlocking fingers, similar in morphology for those described for the frontal-nasal suture.

Dental microwear

Qualitative assessments of dental microwear texture reveal patterns of anisotropy (parallel ridges) and low complexity on the labial margin of the upper left postcanine attributed to a small individual of Exaeretodon argentinus (see Figs. 8A–8C). However, wear facets (Fig. 8D) appear to have little anisotropy and are largely dominated by some degree of pitting and irregular abraded surfaces. The only other account of dental microwear texture in E. argentinus focused on what were considered to be adult specimens (PVSJ 707: BSL∼29 cm; PVSJ 1091; BSL∼30 cm; Kubo, Yamada & Kubo, 2017). Microwear in purported adult specimens revealed primarily anisotropic patterns, wherein the majority of scratches are oriented within 20°  of the anteroposterior orientation of the tooth (Kubo, Yamada & Kubo, 2017). The patterns evidenced herein appear to be oriented more along the dorsoventral axis of the tooth and are thus ∼70°  displaced from scratches on teeth of larger individuals. Microwear displacement between small and large individuals may reflect masticatory and/or dietary differentiation between small and large individuals.

Figure 8 SEM images of a tooth cast, capturing the labial face of an isolated postcanine tooth attributed to MCZ VPRA-4470.

Central panel, line drawings of the upper left postcanine in distal and labial views (from top to bottom). Grey boxes are an approximate location where the images are taken from. (A) Groove between primary and secondary cusps (mesialmost), in the apicobasal midpoint of the cast; (B) labialmost portion of the primary cusp; (C) groove between primary and secondary cusps, just below the apical margin; (D) area surrounding wear-facet (bottom left), located at the apical tip of the primary cusp. Arrows denote the mesial (anterior) direction, with striations (anisotropy) perpendicular to the mesiodistal axis of the tooth. Dark rounded areas are holes and are an artifact of bubbles in the casting material. SEM scale bar equals 1 mm.

Discussion

Osteological shifts through ontogeny

Discerning age and overall developmental patterns from the fossil record using only size is a difficult task that requires multiple lines of evidence to best ensure that a growth curve represents only one taxon (Sampson, Ryan & Tanke, 1997; Abdala & Giannini, 2000; Griffin & Nesbitt, 2016; Hone, Farke & Wedel, 2016; Griffin et al., 2021). We assess morphological change by modeling changes in size through allometry, using regressions to hypothesize how a species may have developmentally changed through age. By combining such quantitative change through regressions with qualitative changes in external morphology, we can more appropriately estimate ontogenetic trajectories in extinct taxa. There are striking morphological differences in Exaeretodon argentinus from small (14.9 cm long) to large skulls (49.6 cm long). More than half (nine of 16) of cranial features measured for Exaeretodon argentinus are isometric with respect to differences in basal skull length (BSL; that we interpret as a proxy of age). Most notable deviations from isometry are expansion of the temporal region (TEL), and the overall reduction of the braincase length (BB), orbit length (OL), and the palate length (PAL) and associated upper dentition length (UP). Additionally, under the GLMM, the width of the transverse process of the pterygoid (TP; negative allometry), and the height of the zygomatic arch (ZH; positive allometry) are marginally significant. The lateral face of the transverse process is in contact with the medial surface of the mandible (Crompton, 1995) and with the reduction in width of the transverse processes (TP), we expect a relatively more central position of the mandible in the temporal opening, which likely indicates more developed occlusal musculature. A similar trend, although more extreme, was reported for the epicynodont Galesaurus, in which the mandible of younger individuals was laterally located near the zygomatic arch, indicating widely separated transverse processes (Jasinoski & Abdala, 2017a).

Myological shifts through ontogeny interpreted through allometric relationships

Evaluating changes in musculature based on allometric coefficients often requires an assumption that measurements on a dry skull (surface area of attachment sites) are roughly proportional to muscle dimensions measured from a dissection. A dry-skull inference will always be an underestimate of muscle size, as the smallest a muscle can ever be is the osteological area of attachment. As it stands, the dry skull inference may not be useful or accurate when comparing multiple species to one another, even those from the same clade (Law & Mehta, 2019; Bates et al., 2021). Surprisingly, given an ontogenetic dataset, size and covariation between muscles and their attachment sites, or even between different attachment sites, may also not be reflective of the musculature or functional estimates derived from musculature (CJ Law pers. comm, 2021; Law & Mehta, 2019). Critically, using only the measurement of an attachment site in an individual mammal, or a growth series of a species of mammals, is not a reliable method to interpret function or ecology (Hutchinson, 2012; Toro-Ibacache, Muñoz & O’higgins, 2015; Bates et al., 2021; Broyde et al., 2021). As such, we looked to combining changes in allometric parameters (slope and intercept), to evaluate the rate of change between features, with overall morphology and how their correlations may be reflective of function and ecology.

The m. masseter and temporalis are often discussed in their roles in chewing and crushing, respectively. The m. masseter anchors onto the zygomatic arch on the skull, and, the masseteric fossa on the coronoid process of the dentary (Crompton, 1963; Lautenschlager et al., 2017). The bounds for the m. masseter in E. argentinus are estimated herein based on the zygoma height (ZH) and width (ZW). The m. temporalis anchors to the sagittal crest and occupies the available space in the temporal fenestra, and its bounds are estimated herein based on the length of the temporal region (TEL). We refrain from assigning skull width (SW) as a proxy for m. temporalis volume, as this space also includes the coronoid process of the dentary and associated musculature. Thus, we focus on length of the temporal region (TEL), but include discussions of skull width, as it can be interpreted as an overall proxy for feeding muscle size. The width of the skull (SW) and width of the zygomatic arches (ZW) have overall isometric relationships, indicating a relatively constant relationship as skull length increases. Conversely, the length of the temporal region (TEL), and height of the zygoma (ZW) have positive allometric relationships, indicating a faster rate of growth in comparison to skull length. Notably, the m. masseter originates in smaller individuals of E. argentinus at a relatively smaller size (ZH; β= −1.33) than the m. temporalis (TEL; β= −0.83), suggesting a greater reliance on the m. temporalis in neonates. The rate of change in the m. masseter (ZH; α = 1.31) outpaces that of the m. temporalis (TEL; α = 1.18), suggesting that the m. masseter could be growing at a faster rate than the m. temporalis. Furthermore, the presence of highly interdigitated sutures between the frontals and the nasals in larger individuals may reflect a structural counterbalance to the strain imposed by the m. masseter as well as the m. temporalis (Herring & Teng, 2000). It is possible that the suborbital process of the jugal (attachment site for superficial m. masseter) also has a positive allometric relationship with skull length (see Fig. 5); however, this feature was not explicitly measured here, as the suborbital process of the jugal extends ventrolaterally and is thus difficult to measure in photographs that are in dorsal or lateral views. Additionally, the suborbital process of the jugal is a feature not ubiquitous to all cynodonts and would not be readily expanded to broader taxonomic studies including non-cynognathian cynodonts.

Differential rates of change between masticatory muscles would then suggest that Exaeretodon argentinus shifted their masticatory use through growth, with a m. temporalis-driven bite in juveniles, followed later by a m. masseter-driven bite in older individuals. Changes in the relative size of the feeding musculature is likely correlated with diet, such that E. argentinus may have transitioned from a faunivorous juvenile stage to an herbivorous adult stage, as is seen in Australian skinks (e.g., Duffield & Bull, 1998). Further investigation into stable isotope geochemistry (e.g., isotopic carbon/nitrogen as proxies for trophic position) and long-bone histology (e.g., lines of arrested growth and vasculature as proxies for continuous or periodic growth) will be imperative to support this hypothesis in future studies. Presently, we explore observations of sutural complexity and dental microwear between small and large individuals of E. argentinus to evaluate whether additional systems support dietary differentiation through growth.

Changes in suture complexity over the Exaeretodon growth series

Sutural contacts between cranial bones allow for individual bones to grow dynamically throughout ontogeny (Di Ieva et al., 2013). Sutural shape and complexity varies widely across vertebrates, as well as in relation to ontogeny and muscle actions (e.g., pig, mouse, and theropods: Herring & Teng, 2000; Byron et al., 2004; Rayfield, 2005), and to behaviors such as head-butting (Nicolay & Vaders, 2006; Di Ieva et al., 2013; Benoit et al., 2017). Structural differences between sutures are commonly discussed as simple—sutures appearing mostly straight on the surface of the bone—or complex sutures that show varying degrees of interdigitation (i.e., notched contacts where finger-like projections of each bone overlap one another to increase surface area of contact). However, these sutures are not always rigid but allow for some degree of flexion between bones as well as acting as reservoirs for mechanical stresses (Herring & Teng, 2000; Di Ieva et al., 2013). Notably, the contraction of the m. masseter and m. temporalis in mammals have been shown to propagate as tensile forces in pigs, primarily in the anterior region of the interfrontal suture, and the posterior region of the interparietal suture, respectively (Herring & Teng, 2000). It would thus follow that greater tensile forces would correlate with greater degrees of sutural interdigitation amongst the frontals and parietals, and the surrounding bones, to offset greater strain magnitude or duration. Sutural complexity has been shown to be correlated with both muscle mass and food toughness in euarchontan mammals (Byron et al., 2004; Byron, 2009). Taking into account these evidences, we deduce that the appearance of surface interdigitation throughout ontogeny would reflect an increase in muscle mass/activity, and that the location of such interdigitation may be reflective of redistributing masticatory strain to adjoining bones. From this, we hypothesize that the presence of increased sutural complexity surrounding the anterior extent of the frontals to be indicative of an increased contribution of the m. masseter, and that m. temporalis-dominated mandibular movements would produce increased sutural complexity in the posterior margin of the parietals and adjoining bones. To the best of our knowledge, the hypothesis that increased contribution of the m. masseter correlates with increased sutural complexity amongst the frontals has not been formally tested, and so we view these results as being consistent with the current literature, but they are the least influential in estimating ontogenetic ecology in Exaeretodon argentinus.

In Exaeretodon, larger individuals bear more interdigitating sutures in the skull (e.g., frontal-parietal vs parietal-parietal sutures), throughout the snout and braincase, reflecting enhanced structural integrity for increased and sustained force exertion during mastication, and the ability to process harder (e.g., more fibrous) food objects (Monteiro & Lessa, 2000; Byron et al., 2004; Byron, 2009). The snout and braincase are regions where multiple bones contact one another resulting in highly sutured areas. Increases in complexity (i.e., interdigitation vs simple contacts) through growth reflects an increase in surface area of the sutural contacts, which suggests that the snout and braincase would be more resistant to stresses (such as masticatory), in older individuals (White et al., 2020). This is further supported by the lack of deformation (e.g., element displacement) in the snout and braincase of larger individuals. The process of increasing sutural complexity does not include the maxilla, as it retains a simple suture throughout this growth series, and is offset considerably in smaller individuals (e.g., MCZ VPRA-338-58M), as opposed to the condition in some larger individuals (e.g., MCZ VPRA-4483), where maxillary displacement from the nasals is minimal. In our largest individuals (e.g., MCZ VPRA-4483), the degree and overall surface area of interdigitation between the sutures of parietals, squamosals, and occipital bones do not match those of the frontonasals in E. argentinus, which possess longer and more numerous projections per unit length. Given the assumptions discussed above, we interpret this difference in sutural complexity to reflect a masticatory style wherein the m. masseter is the dominant feeding muscle, and thus, more robust sutures are needed to spread greater stresses in the anterior margins of the skull. A difference in degree of sutural complexity and element displacement between small and large individuals may suggest that the snouts are somewhat resilient to forces and stresses that would come from mastication (Jasinoski, Rayfield & Chinsamy, 2010; Maloul et al., 2014) and thus it is key to evaluate and reconstruct developmental patterns for the muscles that dominate feeding in cynodonts, the m. masseter and temporalis. Jasinoski, Abdala & Fernandez (2015) also reported an increased complexity of particular sutures (i.e., nasal-frontal) towards adulthood in Thrinaxodon.

Microwear texture supports heterogeneity in diet through growth

To further evaluate ecological differentiation through growth in Exaeretodon argentinus, we evaluate microwear texture on the labial margin of an isolated upper left postcanine tooth attributed to one of the smaller individuals in our sample (MCZ VPRA-4470; BSL = 16.7 cm). Anisotropy, parallel lines, is the primary texture on the labial surface of the tooth; however, such lines are sparse and are only weakly scraped. These patterns do not reflect scratching or gouging from occlusion during mastication, but instead reflect abrasion as differing food items would have interacted with the labial margin of a tooth throughout mastication (Calandra & Merceron, 2016). Patterns of anisotropy have largely been present in amniotes that have primarily faunivorous diets (DeSantis, 2016; Bestwick et al., 2020). Anisotropy is also recovered as the predominant texture in larger individuals of E. argentinus (Kubo, Yamada & Kubo, 2017), but are instead displaced 70°  in the tooth of the smaller individual (MCZ VPRA-4470). The microwear of larger individuals has been interpreted to reflect propalinal movement of the jaw, which can be interpreted as an m. masseter driven mastication (Kubo, Yamada & Kubo, 2017). Herein, we find that the anisotropic patterns of a small individual (MCZ VPRA-4470) are oriented apicobasally, nearly perpendicular to the orientation of scratches evidenced in larger individuals of E. argentinus (Kubo, Yamada & Kubo, 2017). If the patterns of anisotropy in MCZ VPRA-4470 are similarly reflective of jaw motion, mastication would be primarily orthal, suggesting an m. temporalis driven bite (see Grossnickle et al., 2022 and references therein), contra to the inferred palinal movements in larger Exaeretodon (Kubo, Yamada & Kubo, 2017). Interpreting ecological differentiation based on microwear alone assumes all dental modifications are feeding-related but finding consistent patterns in the ontogeny of masticatory muscle sizes and sutural complexity all together support younger individuals of E. argentinus having a dietary ecology distinct from older conspecifics.

Dietary differentiation through ontogeny in Exaeretodon argentinus

Our hypothesis that Exaeretodon argentinus went through a dietary shift through growth is supported by: (1) relative expansion of the areas and openings for the attachment of the m. masseter and temporalis; (2) increases in sutural complexity in the snout and the cranium; and (3) displacement in microwear anisotropy from small to large individuals. The allometric relationships provide the most convincing evidence of differentiation of the three, whereas sutural complexity and microwear even when providing some support, are not individually compelling. However, taken together, these concurrent changes suggest differentiation in feeding ecology with growth. The shift in the orientation of anisotropy in the microwear suggests that there was a reorientation in the direction of wear produced by food materials, which we interpret as a functional result from a change in chewing motion, related to dietary differentiation toward herbivory that likely occurred between the two sampled size classes (MCZ VPRA-4470: BSL = 16.7 cm vs PVSJ 1091: BSL∼30 cm). It is possible that Exaeretodon was herbivorous throughout life, where the presence of anisotropy could indicate some retention of diet between size classes of between size classes, and that shifts in dominant feeding muscles occurs in tandem with a transition from a more generalist herbivorous lifestyle in smaller individuals to a more specialized diet in larger individuals; however, additional analyses including stable isotope will be necessary to further explore this.

Current assumptions of herbivory in species of Exaeretodon (Martínez et al., 2012; Francischini, Dentzien-Dias & Schultz, 2018; Melo et al., 2019) do not fully capture nuanced morphological changes through ontogeny, which suggests that younger individuals would have been ecologically distinct, and were likely more faunivorous than older contemporaries an ecological strategy known in Australian skinks, bearded dragons, spiny tailed-iguanas, and numerous omnivorous lizards (Duffield & Bull, 1998; Cooper & Vitt, 2002; Wotherspoon & Burgin, 2016). Because we cannot estimate the complete growth model of the taxon, we can’t pinpoint when this transition may have occurred, but we can recognize that younger (smaller) individuals are morphologically, functionally and ecologically distinct from older (larger) individuals.

Exaeretodon ontogeny suggests similar development amongst cynognathians

The proposed changes in dietary ecology through growth in Exaeretodon argentinus parallels macroevolutionary patterns of dietary change across Cynognathia (see Fig. 9). Based on recent phylogenetic hypotheses of cynodont relationships (e.g., Lukic-Walther et al., 2019), the plesiomorphic diet for Cynognathia is reconstructed as primarily carnivorous, with omnivory proposed at the base of Gomphodontia (see Botha, Lee-Thorp & Chinsamy, 2005), and strict herbivory appears in the base of the Traversodontidae, with the exclusion of Etjoia dentitransitus (Hendrickx et al., 2020). Parallels between the ontogeny of E. argentinus and the evolution of diet across Cynognathia may indicate some degree of developmental similarity that is revealed via heterochronic shifts (see Klingenberg, 1998) towards larger body size in the more well-nested Gomphodontosuchinae.

Figure 9 Phylogeny of cynodont relationships based on Lukic-Walther et al. (2019), with the inclusion of Siriusgnathus, and gomphodontosuchine relationships based on Hendrickx et al. (2020).

Species that have had cranial ontogenetic studies are highlighted in pink. Exaeretodon argentinus is represented in bold and with an asterisk. Steak and leaf icons represent faunivory (carnivory and/or insectivory) and herbivory, respectively. Steak on leaf icon represents omnivory. Question marks indicate uncertainty in diet. Dietary assumptions are based on discussions in various literature (Crompton, 1995; Liu & Abdala, 2014; Hendrickx et al., 2020). Uncertainty in the diet of Siriusgnathus and Exaeretodon riograndensis is represented because of their close affinity to Exaeretodon argentinus, but their absence in this current study.

We assume that some degree of postdisplacement and/or acceleration is evident in Exaeretodon, as existing allometric differences appear to largely relate to showing increased growth rates (Klingenberg, 1998). The allometric coefficients, the estimated size at birth for the taxon (y-intercept), as well as some features of Exaeretodon argentinus are consistent with those of Massetognathus pascuali (Abdala & Giannini, 2000), based on shared patterns of positive allometry for the length of the temporal region and zygoma height, and negative allometry for the length of the palate, orbit, upper postcanine series and braincase. Although, these taxa differ in the magnitude of estimated slopes (e.g., rates of change) and significance of those slopes, conserved patterns of positive and negative allometry suggest a similar growth pattern between these well-nested traversodontids (Table 4). However, a larger y-intercept suggests that E. argentinus would have had larger neonatal offspring than M. pascuali, and thus energetic demands for growth would differ between these taxa. Furthermore, the relative expansion and reduction of the zygoma height and length of the upper postcanine series, are consistent with patterns of the ontogenetic series of Diademodon tetragonus (Bradu & Grine, 1979; Grine & Hahn, 1978; Grine, Hahn & Gow, 1978), one of the earliest diverging cynognathians, potentially indicating that some aspect of cranial development is conserved throughout cynognathians and possibly even some non-eucynodont epicynodonts. Recovering a consistent allometric pattern across sampled Gomphodontia—the last common ancestor of Diademodon tetragonus and Exaeretodon argentinus and all of its descendants (Sues & Hopson, 2010)—suggests that the evolution of diet across Cynognathia may be a consequence of conserved cranial development, and that we may need to investigate postcranial remains to better understand ecological differentiation amongst these animals.

Known cynognathian allometries reflect a consistent pattern within Gomphodontia of the zygoma outpacing the relative growth of the temporal region (Table 4). Such a relationship is shared in the early-diverging probainognathian, Chiniquodon theotonicus (Abdala & Giannini, 2002), which is presumed to be carnivorous throughout life. However, a contrasting pattern is shown in the early-diverging epicynodonts, Galesaurus planiceps and Thrinaxodon liorhinus, which show a relatively faster growth for the temporal region over the height of the zygomatic arches. Taken together, these patterns suggest that there may have been a common developmental trend for eucynodonts, which is distinct from their earlier diverging relatives, suggesting that the patterns seen in Exaeretodon are likely not plesiomorphic to Cynodontia. Notably, allometric patterns for Exaeretodon, do not follow patterns of positive craniofacial allometry (cranial rule of evolutionary allometry: see Cardini, 2019), but instead show isometric growth in the snout, consistent with current hypotheses that a lack of the “cranial rule of evolutionary allometry” pattern may be plesiomorphic for the earliest mammals (Krone, Kammerer & Angielczyk, 2019); however, future studies that explicitly incorporate phylogenetic structure will be necessary to assess the origin and evolution of the “cranial rule of evolutionary allometry”. The data presented herein for Exaeretodon and other non-mammalian cynodonts show that taxa inferred to be faunivorous (e.g., Galesaurus, Thrinaxodon, and Chiniquodon), do not show consistent morphological ontogenetic patterns with one another. This suggests that the link between diet and ontogeny are nuanced, and that multiple lines of evidence may be necessary to reconstruct diet because other features of cranial development may be important for dietary differentiation between species.

Until recently, the Traversodontidae has been considered a clade of likely exclusive herbivores, although the discovery of Etjoia dentitransitus as one of the earliest-diverging traversodontid suggests omnivory as a possible plesiomorphic condition for the clade (Hendrickx et al., 2020). By interpreting young Exaeretodon as an omnivorous traversodontid, it would follow that omnivory may not have occurred within the Gomphodontosuchinae, but as a heterochronic shift toward larger body size in neonates. Additional analyses of ontogeny in traversodontids will be critical to assess if faunivory in young individuals is consistent across the entire clade, or if Exaeretodon represents a relatively slow development, in which masticatory modification occurs postnatally. Regardless, the inclusion of animal material in diets amongst well-nested herbivores indicates a need to re-evaluate the evolution of herbivory across the Traversodontidae, and how differing developmental strategies are associated with ecological differentiation.

Supplemental Information

Supplemental Information 1 Supplemental document

Click here for additional data file.

Supplemental Information 2 Raw mm measurements of features

Click here for additional data file.

Supplemental Information 3 Associated code

Click here for additional data file.

Supplemental Information 4 Log-transformed data

Click here for additional data file.

The authors thank Stephanie J. Pierce, Jessica Cundiff, and Christina Byrd at the Museum of Comparative Zoology, Harvard University for allowing access to specimens. They acknowledge that all Exaeretodon from the Harvard Museum of Comparative Zoology were collected from Argentina in the 1960’s by Alfred Sherwood Romer and colleagues, and rightfully belong to the province of San Juan, Argentina, following amendments to the Argentine constitution in 1994, and the subsequent Archaeological and Paleontological Heritage Act in 2003. They thank Christian Kammerer, for providing photographs of various cynodontians that made this work possible, given travel restrictions due to COVID-19. They thank Chris J. Law for discussion of dry skull muscle estimation and providing data to run additional tests. They thank the handling editor, Diogo Provete for their patience and assistance in this process, as well as Leonardo Kerber, and two anonymous reviewers whose comments greatly improved the quality of these works. They also thank Josef Uyeda, Michelle Stocker, Christopher Griffin, Casey Holliday, and the Virginia Tech Paleobiology group for helpful discussion.

Institutional Abbreviations

MACN Museo de Ciencias Naturales “Bernardino Rivadavia”, Buenos Aires, Argentina

MBR Museum für Naturkunde Berlin, Berlin, Germany

MCZ VPRA The Louis Agassiz Museum of Comparative Zoology, Cambridge, Massachusetts, USA

MLP Museo de La Plata, La Plata, Argentina

PVL Colección Paleontología de Vertebrados Lillo, Universidad Nacional de Tucumán San Miguel de Tucumán, Argentina

PVSJ Museo de Ciencias Naturales, Universidad Nacional de San Juan, San Juan, Argentina.

Additional Information and Declarations

Competing Interests

Author Contributions

Data Availability

The authors declare there are no competing interests.

Brenen Wynd conceived and designed the experiments, performed the experiments, analyzed the data, prepared figures and/or tables, authored or reviewed drafts of the article, and approved the final draft.

Fernando Abdala conceived and designed the experiments, performed the experiments, authored or reviewed drafts of the article, and approved the final draft.

Sterling J. Nesbitt conceived and designed the experiments, authored or reviewed drafts of the article, and approved the final draft.

The following information was supplied regarding data availability:

The raw data, and log-transformed data for the analyses, as well as the associated code, are available in the Supplementary Files.

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
