# Peer review of "Ontogenetic growth in the crania of Exaeretodon argentinus (Synapsida: Cynodontia) captures a dietary shift"

_PeerJ, doi:10.7717/peerj.14196_

## Round 0.1 · original submission · Major Revisions

Your manuscript has now been assessed by two independent reviewers. Notice R1 makes many comments on key aspects of the manuscript, especially about the interpretation of the results. (S)he also provides alternative interpretation that authors should at least consider. This reviewer also points to the need to make broader comparisons across related groups. I also second his/her comments about the need for a figure that shows the slope differences.

The information in Table 2 is replicated in Fig 1. There's no need to show both in the main text, just stick with Fig 1. Table 1 should be a supplementary material.

Legends of Figs 5 and 6 are switched.

You certainly need to actually compare the slopes, as the reviewer also points out. The fact that you have used traditional linear morphometrics instead of geometric morphometrics complicates a bit not only this aspect, but also the issue with the representation (s)he mentions. With geometric morphometrics you could also get rid of the deformation more easily. However, there's still a way to work this out. Adams & Nistri (2010) have proposed a better way to visualize multivariate allometry.

https://link.springer.com/article/10.1186/1471-2148-10-216

Another point: Given that you’re not describing the species for the first time nor have any new material added, I’m not sure if it makes sense to have a ‘Systematic paleontology’ section in the manuscript.

I also don’t see the need to have a lengthy section ‘Contemporary methods to assess ontogenetic age in non-mammalian synapsids’ in the Methods. It’s not your goal do review the methods and I’m really not sure if this actually adds something relevant to the manuscript. Overall, your manuscript is much longer than it needs to be. Authors have already explained their method adequately in 275-301. And here comes my main question: why have you considered each bone/feature separately and conducted a GLMM, instead of using multivariate techniques that are more adequate to your case? I know the same authors have proposed this method in a recent paper, but a better explanation of why you have chosen univariate instead of multivariate methods would be fair.

I can see authors are very reluctant in classifying specimens as an adult or juvenile due to a lack of proper data. However, an allometric study doesn’t necessarily need this discrete classification, yours certainly don’t.

Finally, PeerJ uses a structured abstract. I highly recommend you adhere to it when preparing the revised version.

Reviewer 1 ·

Excellent Review

This review has been rated excellent by staff (in the top 15% of reviews)
EDITOR COMMENT
The reviewer has done a pretty good job summarizing the big strengths of the paper, while also providing specific guidance as how to improve less structured aspects of it. One of their major contributions was bringing up alternative explanations for the data found, which certainly will help authors improve the paper.

Basic reporting

Good. Some typos, so please carefully proofread. Raw data is included. See below for specific comments.

Experimental design

Good. See below for specific comments.

Validity of the findings

Some findings need more qualification. See below for specific comments.

Additional comments

The authors examined skull ontogeny in the cynodont Exaeretodon argentinus. As expected, they found that there are positive allometries associated with the regions of the skull that contain the masticatory muscles. Per the authors, the linear measurements that correspond to these regions are the height and width of the zygomatic arches (ZH and ZW), the width of the skull (SW), and the length of the temporal region (TEL). The authors take ZH and ZW together to represent the volume of the masseter, and SW and TEL together to represent the volume of the temporalis. ZH and TEL have positive allometries with respect to skull length, and ZW and SW are isometric. The authors interpret a small difference in allometric slope between ZH (slope = 1.31) and TEL (slope = 1.18) as evidence that the masseter is growing faster than the temporalis; they further suggest that younger, temporalis-dominated individuals may have been faunivorous, as opposed to older, masseter-dominated individuals, which are inferred to have been herbivorous. They interpret this ecological shift through the lenses of ontogeny and (somewhat confusingly) phylogeny. They support this faunivorous-to-herbivorous scenario with qualitative evidence based on sutural complexity and tooth microwear.

I find the authors’ ecological hypothesis interesting and plausible, although not definitive. That the masseter has positive allometry with respect to the temporalis is fairly well demonstrated by their analysis. The inference about food types is a bit shakier, and I would suggest adding some qualifications. For example—I realize this sounds a bit silly—couldn’t the young, temporalis-dominated individuals have been eating plant foods that are soft or otherwise amenable to up-and-down chewing rather than grinding? Young individuals may have been more faunivorous than older individuals, but I don’t think we have nearly enough evidence to state definitively that they are more accurately described as faunivores than as herbivores.

Also, an overarching problem is narrowness of comparative scope: the authors frequently discuss other traversodontids, rarely other cynodonts, and almost never any other vertebrates (except selected, questionably relevant lizards). The authors frame their findings as shedding particular light on Exaeretodon and other herbivorous cynodonts, but I wonder whether their masseter vs. temporalis pattern can be seen broadly in other amniotes (or other vertebrates).

Here are my major suggestions for the authors, with more minor comments at the end:

1) I think you must add relevant outgroup comparisons to most sections of the Discussion. For example, in the paragraph starting on line 637, where you make a functional interpretation of the ZH vs. TEL relationship in Exaeretodon, can you address—even qualitatively—ZH vs. TEL relationships in other amniotes, including non-herbivores? Is there evidence that carnivores ever exhibit faster masseter growth? Is it possible that faster masseter growth is plesiomorphic for cynodonts (or synapsids, or amniotes)? Similarly, the pattern of increasing sutural complexity that you observe seems, to me, basically as expected for any synapsid. Can you make it clearer where you are making a particular statement about Exaeretodon—which would be relevant to your ecological hypothesis—and where you are making more general statements? In addition, can you make it clearer whether you are making general claims about increasing mastication forces with age (relevant to ZH and TEL equally) or claims about masseter-dominated chewing vs. temporalis-dominated chewing (relevant to the ZH vs. TEL comparison)?

2) I agree with you that the temporalis occupies most of the temporal fenestra, but the masseter is also in that space. Can you more clearly justify treating TEL, in particular, as reflecting temporalis but not masseter volume? This should probably go in the paragraph starting at line 619.

3) I’m not prepared to comment in detail on the statistical procedures, but can you explain in the Methods why it is proper to compare the allometric slopes for ZH and TEL, as you have done, without doing some sort of formal statistical comparison test? Similarly, I’m a bit concerned that you establish that ZW and SW are relevant to the jaw muscles, then discard them and compare only ZH and TEL. I understand that neither ZW nor SW has a pattern that is statistically different from isometry, but do their allometric slopes differ from each other? If so, should those differences be considered in the comparison between masseter and temporalis? Perhaps not, but I think you should more clearly justify this choice.

4) I think you need a figure in the main text that clearly shows the difference in slope between ZH and TEL (and ZW and SW?). Ideally, they would be shown on the same plot along with a line of isometry, so that the masseter vs. temporalis pattern is apparent visually.

5) I realize that you are reluctant to establish a sharp boundary between juveniles and adults, but you need to state more clearly at what size (in terms of raw BSL) the crossover from temporalis-dominated to masseter-dominated chewing is predicted to occur. This is important so that your results can be more easily interpreted biologically, and it is relevant to the tooth microwear section: does the shift occur between 16.7 cm (your small individual) and 29/30 cm (the individuals from Kubo et al. 2017)? In other words, do the microwear results accord with the allometry results? Also, I think you need to discuss this explicitly when you make comparisons with herbivorous lizards, which are all presumably much smaller than your young Exaeretodon individuals when they switch from faunivory to herbivory.

6) In the microwear section, can you discuss why you chose the labial surface of the tooth to examine and whether that surface is comparable to the surfaces examined in other microwear studies, especially Kubo et al. 2017? Did you look at other surfaces, and if so, what did you find? I’m a bit concerned that scratches on the labial side rather than the occlusal surface do not reliably reflect diet, so I think you need to justify your inferences more clearly. Also, please explain how you chose this particular tooth: is it the only isolated tooth in your sample? In general, I think you should treat these results as preliminary and suggestive, at most. For examples, lines 705–710 seem to treat the microwear evidence as on par with allometry and sutural complexity, whereas I think the allometry evidence is by far the strongest of the three.

7) I find the section on developmental constraint (starting at line 722) to be confusing. Many of your statements are either difficult to understand or dubious. I would probably delete the entire section.

8) I think you need to address the possibility of sexual dimorphism somewhere in the text. If some of the small individuals are adult females (or I suppose males) rather than juveniles, does that affect your interpretation?

9) “Zygoma” (many places in the manuscript) is singular. For the plural, use “zygomas” or “zygomata” (per OED) or “zygomatic arches.”

10) “Paraphyletic clade” (line 103) is a contradiction in terms.

11) Throughout the Results and Discussion, please be careful to include the p-value every time you discuss whether a variable is isometric or allometric (even though I realize this means repeating some of the p-values many times).

12) Is there a reason that ZW is missing from Table 4?

13) I think you should consider adding a simple phylogeny that includes the taxa in Table 4 and shows some of the clades you discuss frequently (Traversodontidae, Gomphodontia, Cynodontia).

·

Basic reporting

Exaeretodon is one of the most abundant non-mammaliaform cynodonts found in Triassic deposits from South America. Therefore, the present study presented by Wynd and colleagues shows a welcome contribution to the study of the ontogeny of this organism. By using a methodology that minimizes the impact of taphonomy on specimens, the study provides a basis for future analyses on other taxa. The authors found an association between cranial ontogeny and change in the diet pattern of the species. This finding can now be studied by different methodologies. The manuscript is well written, the methods are adequate, and there are interesting findings. I suggest coloring the homologous bones in figure 4 and identifying the bones in figure 5 directly over the image. Additional photos of some of the analyzed specimens evidencing important anatomical features would be welcome for future comparisons.

Experimental design

No comment.

Validity of the findings

No comment.

---

## Round 0.2 · Major Revisions

I have received back the comments by two of the reviewers from the previous round. While R2 is OK with the changes made, R1 made a number of key critiques on the way you interpreted the results in relation to development and phylogeny. I'd kindly ask authors to provide a reply to each of them in a revised version of the manuscript and rebuttal letter.

Reviewer 1 ·

Basic reporting

no comment

Experimental design

see additional comments

Validity of the findings

see additional comments

Additional comments

I appreciate the changes the authors made to the manuscript in response to the reviewers’ and editor’s comments. The phylogeny is especially helpful. I will limit my comments to the most important remaining issues.

1) In the absence of other data, that Exaeretodon, Massetognathus, Diademodon, and Chiniquodon have the same developmental pattern suggests that this developmental pattern is most parsimoniously reconstructed as being ancestral to eucynodonts. (Relatedly, I can see no evidence for the statement at line 760 that development in cynognathians is distinct from development in non-cynognathian eucynodonts.) The authors in their rebuttal say that broad phylogenetic comparisons are outside the scope of their paper, but the point is that they must provide a real argument that the pattern in Exaeretodon and other herbivorous taxa has something to do with these animals’ specific ecological strategies, since naively it appears that the pattern is plesiomorphic and -- especially problematically -- is shared with closely related carnivores such as Chiniquodon. The data from Thrinaxodon and Galesaurus don’t refute this; it is easy to imagine that they share a primitive developmental pattern that was replaced on the line leading to eucynodonts.

2) The authors still have not explained in the text why the pattern of increasing sutural complexity provides any evidence of a shift in dietary strategy. Lifelong faunivores also show increasing interdigitation with age.

3) “Current assumptions of herbivory in species of Exaeretodon (Martínez et al., 2012; Francischini, Dentzien‐Dias & Schultz, 2018; Melo et al., 2019) are incongruent with this work” (line 731). I think this and any similar statements are unfair, since as the authors point out many herbivorous species have a more faunivorous juvenile stage. Better to say that the attempt is to add nuance to reconstructions of diet over the lifespan.

4) See below for specific comments on the developmental section. Again, I think the central argument made in this section, if I understand it correctly, is mistaken; regardless, the section is confusingly written and many statements are unclear.

“The earliest diverging cynognathians are reconstructed as primarily carnivorous, to purported omnivorous gomphodontians, and to the well-nested herbivorous traversodontids.” (line 745)

Avoid discussing phylogenies in this linear way. Traversodontids are nested within gomphodonts.

“Parallels between the ontogeny of E. argentinus and the evolution of diet across Cynognathia may indicate some degree of developmental similarity that is revealed via heterochronic shifts in body size in the Gomphodontosuchinae.” (line 750)

Is the argument that large cynognathians are peramorphic? Peramorphosis, which is the addition of new developmental stages to the end of the ancestral developmental trajectory, results in “developmental exaggeration of…ancestral adult trait[s]” (Plateau and Foth 2020, Nature Communications Biology 3:195). Exaeretodon and other large herbivorous cynognathians have specific dental and cranial adaptations for herbivory that are not merely exaggerations of the traits of carnivores; instead, the developmental trajectories of carnivores and herbivores diverge to produce recognizably distinct adult morphotypes, neither of which is an exaggeration (or a juvenile form) of the other. The fact that increased body size seems to have evolved along with increased levels of herbivory in cynognathians is unsurprising for simple energetic reasons, and is in keeping with patterns across many clades of vertebrates. Nothing is implied about the evolution of development.

“Albeit these taxa differ in the magnitude of estimated slopes (e.g., rates of change) and significance of those slopes, but conserved patterns of positive and negative allometry suggest a similar growth pattern between these well-nested traversodontids (Table 4).” (Line 753)

“Albeit” is used incorrectly here and should be deleted.

“Recovering a consistent allometric pattern across presently sampled gomphodonts—the last common ancestor of Diademodon tetragonus and Exaeretodon argentinus and all of its descendants (Sues & Hopson, 2010)—suggests that cynognathian evolution may be due to conserved developmental strategies.”

What aspects of cynognathian evolution? How can evolution (i.e., change) be caused by conservation (i.e., lack of change)?

“development is not directly canalized in Gomphodontia” (line 770)

Developmental canalization is about robustness of within-species phenotype to differences in genotype and environmental conditions. How does this concept relate to the dissimilarity of a theoretical juvenile Exaeretodon to adult Diademodon?

“However, these hypothetical young individuals had relatively smaller temporal regions (average TEL~66% and SW~70%) and much reduced zygomatic arches (average ZH~34% and ZW~58%), compared to a large D. tetragonus individual (Table 5). Given the proposed omnivorous lifestyle of D. tetragonus (Botha, Lee-Thorp & Chinsamy, 2005), these relationships propose neonate individuals of E. argentinus with potentially similar dietary tendencies, based on the relative contribution of the regions anchoring the m. temporalis (TEL), compared to the relatively smaller contributions of the regions anchoring the m. masseter (ZH and ZW). Such a transition from faunivory to herbivory in Exaeretodon argentinus would thus mirror the evolution of diet in traversodontids (Grine, 1978; Botha, Lee-Thorp & Chinsamy, 2005; Goswami et al., 2005; Kubo, Yamada & Kubo, 2017), indicating a degree of phylogenetic retention along with developmental similarity reflective of the broader clade.” (line 782)

Neonatal individuals with dietary tendencies that are potentially similar to… what? It seems like you are saying that hypothetical neonates are more faunivorous than adult Diademodon individuals? Please explain how this indicates phylogenetic retention (of what?) and developmental similarity among traversodontids.

“Taken together, these patterns suggest that there may have been a common developmental trend for epicynodonts, which is distinct from their earlier diverging relatives, suggesting that the patterns seen in Exaeretodon are likely not plesiomorphic to Cynodontia.” (792)

Right. But what about Eucynodontia?

“Furthermore, these data show that taxa inferred to be faunivorous (Galesaurus, Thrinaxodon, and Chiniquodon), do not show consistent patterns with one another. This suggests that the link between diet and ontogeny are nuanced, and that other features of cranial development may be important to dietary differentiation between species.” (796)

Right. So it is somewhat problematic to rely on one aspect of cranial development for dietary differentiation WITHIN species, too.

“By interpreting young Exaeretodon as a faunivorous traversodontid, it would follow that faunivory may not have occurred convergently within the Gomphodontosuchinae, but as a heterochronic shift toward larger body size in development.” (803)

A trait might be both convergent and the result of heterochrony. Also, I still don’t know what you mean when you say there was a heterochronic shift (see above).

·

Basic reporting

Commented on the first review.

Experimental design

Commented on the first review.

Validity of the findings

Commented on the first review.

Additional comments

My suggestions were incorporated in this version, as well as other questions raised by other reviewers were answered. I suggest only including Lukic-Walther et al., 2019 in the reference list.

---

## Round 0.3 · Major Revisions

I have now heard back from one original reviewer and I also invited a new, independent reviewer to comment on the manuscript. I have to say that they continue to express concern about the interpretation of the ontogenetic series and comparison between juvenile Exaeretodon and adult Diademodon, and subsequent ecological interpretation of skull size and dietary shift throughout ontogeny. I'm giving the authors another chance to address these criticisms and resubmit a revised version. I still would like to see this eventually published, but the authors need to do a better job in responding to these concerns.

Reviewer 1 ·

Basic reporting

see additional comments

Experimental design

see additional comments

Validity of the findings

see additional comments

Additional comments

The authors clarified some of their phylogenetic statements and made significant improvements to the section on sutural complexity. However, the arguments made in the developmental section are still extremely unclear. For example, “progenetic predisposition” is a term that does not appear in Klingenberg (1998) or anywhere else in the literature. I assume the authors meant “predisplacement” instead of “predisposition,” but it is not clear to me that that phenomenon is evident in their data. Also, the authors appear to have the concept of “paedomorphosis” (line 813) completely backward. Pedomorphosis means that an adult descendant looks like a juvenile ancestor; the authors test the hypothesis instead that a juvenile descendant (Exaeretodon) looks like an adult ancestor (Diademodon). And I still do not understand the authors’ use of the term “canalization” (e.g., line 815), which as I pointed out in my previous review generally refers to conserved patterns of development WITHIN species.

Reviewer 3 ·

Basic reporting

See below

Experimental design

see below

Validity of the findings

see below

Additional comments

General comments: This paper uses a combination of morphometric analyses of skull proportions and considerations of some more discrete aspects of skull moprhology (like sutural complexity) to examine whether a dietary shift occurred over ontogeny in the cynodont Exaeretodon. The authors find some evidence in each of these areas, as well as from dental microwear, that suggest differences in muscle activity, chewing, and potentially the resistance of food items that potentially would be consistent with such a shift. That portion of the paper mostly makes sense to me. The paper is also interesting in trying to take the effects of deformation into account in the morphometric analyses.

However, the authors then attempt to use comparisons with more basal cynodonts to argue that the proposed ontogenetic dietary shift in Exaeretodon recapitulates an evolutionary dietary shift within Gomphodontia, but their argument lost me. I think the argument they are trying to make is basically 1) juvenile Exaeretodon resemble adult Diademodon in skull proportions; 2) adult Diademodon were omnivorous so juvenile Exaeretodon were too (on the basis of similar skull proportions); and 3) the evidence for a dietary shift over ontogeny in Exaeretodon suggests that they moved toward a more herbivorous diet over ontogeny, following the evolutionary trend in Gomphodontia towards more herrbivory. However, it seems like their juvenile Exaeretodon don’t show the same skull proportions as as adult Diademodon, so that part of their argument doesn’t seem to hold up. I think it would be useful to include some more extensive comparisons of skull proportions across ontogeny in both Diademodon and Exaeretodon. That would presumably provide more insight into how much they resemble each other at different ontogenetic stages, and therefore when similar diets might be expected, as well as the degree to which ontogenetic patterns are conserved.

In addition to this problem, I think some reorganization of the introduction and methods section might result in a more logical presentation of information (see comments below), and I thought Fig. 5 could do a better job of showing some of the differences that it is supposed to portray.


Line 29: delete comma after Late Triassic

Line 50: I found this statement to be a little confusing because it was unclear to me whether you meant this as a generalization about traversodonts or gomphodonts, or if you intended it as a generalization for most/all herbivorous tetrapod clades.

Line 69: I suggest a bit or reorganization here. I think you should move the paragraph currently starting at line 82 up so that it is the second paragraph in the introduction. That way you have a more consistent discussion of allometry in extant and fossil taxa. Then in the third paragraph, you can bring in the issue of ecology, the observations in modern taxa, and the difficulties of similar studies in the fossil record. As part of that, the sentence that currently starts at line 94 should be incorporated into the ‘ecology’ paragraph (or potentially can be deleted because it seems a little redundant with things that are already in that paragraph).

Line 72-74: These concerns might be semantic, but a couple comments on the statements here. First, insectivory strikes me as a subset of faunivory (i.e., the latter implies eating animals in general, which would include insects). Alligators would seem to become more carnivorous (i.e., focusing on vertebrate prey) and durophagous as they get larger. Second, altriciality in (some) birds and mammals seems to extend well past diet and being fed by adults, and conversely even precocious mammals like most ungulates still rely on milk from their mothers even if they are capable in engaging in other more adult-like behaviors and locomotion. I’m not sure if there’s a better term for it, but it might be good to note more specifically that you’re discussing being fed by parents (often with a specific food source produced by the parent) here.

Line 97: and extant species?

Line 100: replace ecological with ecology or ecological change

line 107: I don’t think cynodontian is used in this way very often. I recommend using cynodont instead.

Line 113: change ‘more specifically’ to especially

Line 120: change to “...reached relatively large body sizes and is known from tens of...”

line 124: delete comma after traversodontid

Line 163: change to physical specimens and photographs

Line 182: Given that defomation is a fairly important consideration in this study, I think it would be useful to have a figure that shows some example specimens that display common or stereotypical forms of deformation (similar to Figs. 2-6 of Kammerer et al. 2020).

Line 183. I think this paragraph should be moved up so that it is the second paragraph in the data collection section. It makes more sense to me to describe the measurements first, and then discuss distortion and how you quantified the effects of distortion on specific measurements. Also see my comment below for line 207.

Line 202: change to: “...do not follow individual...”

Line 205: Are there any discrete characters (either that you have observed or that have been suggested previously in the literature) that potentially could be sexually dimorphic? If not, this might be something that is useful to mention here as additional support for your lack of sexual dimorphism result.

Line 207: I think that this paragraph should be moved up so that it is part of the paragraph (alsoo dealing with distortion) that currently starts at line 174. That way all of your discussion of distortion (and your measurements of it) will be together in one place.

Line 218: I’m not sure I would treat damage like you describe for the premaxilla as a case of distortion. It seems qualitatively different, and like it might affect the resulting measurement differently.

Line 227: This section seems like a bit of a digression in its current location. I think the first paragraph of the section might be better in the introduction of the paper, perhaps towards the beginning, where it can help to explain why you’re looking at allometric patterns. The second paragraph might fit better in the ‘specimens’ section of the methods, since it includes information about the nature of your sample (e.g., that it doesn’t include specimens that are less than about 30% of maximum size) and how you treat them (e.g., not explicitly identifying juvenile classes).

Line 229. add a period after truth

Line 293: Earlier in the paper, some of your comments make it sound like patterns of distortion are somewhat stereotyped, which seems relevant here. So, I recommend reminding readers of the fact. Also, if you include a figure showing examples of deformed specimens, itwould be good to call that our here.

Line 297: suggest instead of suggests

Line 333: Why was that Diademodon specimen chosen for comparison?

Line 349: I recommend noting the basal skull length of MCZ VPRA-4470 and its percentage of maximum size. Also, I think you should mention that you compared the microwear results to the observations in the Kubo et al. paper you cite later in the manuscript. That way readers won’t be left wondering why you didn’t sample an adult specimen for comparison.

Line 373: delete extra period

Line 402: do any of these specimens include a mandible to help you link them back to the holotype? If they do, that would be good to mention here, because it would strengthen the position of these specimens in the chain of evidence underlying the referral of other specimens to E. argentinus.

Line 422: replace thinner with narrower

Line 438: I recommend adding a sentence to the end of this section that calls out figure 4, because it’s a graphic summary of the proportional differences your allometric equations are detecting.

Fig 4. I think this is a very useful and interesting figure. However, since you’re not trying to show size exactly in the figure, I recommend making the skulls the same size (presumably by using the same basal skull length for both the small and large specimens). I think that would better emphasize the proportional changes that you’re using the figure to show. If you want to include some information about the relative sizes of the specimens, maybe you could include a silhouette showing the size of the smaller specimen in each panel, similar to the silhoutettes used in fig. 3 of Krone et al. (2019).

Fig.5: A couple of suggestions here. First, I recommend using bone names or abbreviations consistently instead of a mix of both. Given space constraints, probably only abbreviations would be the better choice. Second, some of the photos seem kind of dull compared to the birght white overlay. Can you increase the brightness of them at all? I think that would help to make some of the details in the photos easier to see and to compare with the overlays (e.g., the nasofrontal suture on the right side of the specimen in Fig. 5b).

line 471: change to contacts

Line 477-483: I’m confused by this section, and I don’t really feel like it is shown well in the figures (which may be the source of my confusion). The difference in anterior extent of the frontal is pretty subtle in Fig 5, but I feel like the front extends a bit farther forward, and the nasal farther psoteriorly, in the larger specimen (5B). This also seem to be implied in Fig. 4. The frontal seems to extend anteriorly between the nasals to a greater degree there as well. Did you get the small-large polarity of the change backwards in the text description?

Line 485: since you list the other bones individually, I recommend using maxilla instead of maxillae

Line 494: call out table 4

Line 516: Change to something like: “Reconstructions of the cranial musculature of both extinct and extant synapsids...” The current wording sounds a bit like the reconstructions themselves are extinct or extant.

Line 521: Do the confidence intervals for the allometric coefficients of temproal length and zygomatic height overlap at all? If they don’t that would provide some additional support for the assertion that they were changing at different rates.

Line 531: I recommend making the section of this paragraph dealing with the nature of the sutures contacts into a separate paragraph.

Line 533: Fig. 5E doesn’t show this well. It seems to focus on the suture between the lacrimal and prefontal. Although that suture is indeed interdigitated, it’s not really what I would consider part of the temporal region, and those elements also aren’t included in the list that you give at the start of the temporal region discussion.

Line 536-538: Are the figure callouts correct here? There’s only one suture highlighted in Fig. 5D, but you seem to be using it as an example of both a less interlocking and more interlocking suture. The use of Fig. 5E here is problematic for the same reasons as I mentioned in my previous comment.

Line 542: I think you should call out Fig. 5F here.

Line 556: Do you think this is related to the isometry observed for squamosal width and zygomatic width?

Line 649: Fig. 4 makes it seem like the suborbital process gets larger in larger specimens. It doesn’t seem to be something you measured, but do you have nay evidence of it being positive alllometric? Considering that it is often considered to be an attachment site for the superficial masseter, it might provide additional support for your hypothesis of increased use of the masster in larger individuals.

Line 669: allow instead of allows

ling 699: change to: “...literature, but they are the least...”

Line 727: The masseter is generally not thought to be present in therapsids other than cynodonts (although dicynodonts likely had something of a masseter analog).

Line 765: delete on

Line 767: I think you statement about a shift toward herbivory based on microwear needs more explanation or justification. In the previous microwear section, you noted that anisotropic scratches such as you observed in both the juvenile and adult teeth are usually associated with faunivory. So, this would suggest that the change in orientation of the scratches (and the presumed change in chewing motion)are indicative of herbivory, but it would be good to state why you think this is the case as well as why you think it didn’t result is a change in the type of wear that was happening.

Line 791: I think you should state a new paragraph with the sentence that begins with ‘the assumptions’.

Line 803: I can see why you say Exaeretodon and Massetognathus are similar in allometric patterns, but Diademodon shares a lot fewer similarities; indeed, Thrinaxodon and Galesaurus technically share more allometric patterns with Exaeretodon than Diademodon does (although some data are missing for Diademodon). So, you might need to temper your statements about shared developmental patterns in Diademodon, or expand them so that it the pattern is share more widely, and perhaps has its origin among basal cynodonts.

Line 811: I recommend starting a new paragraph here.

Line 813: I’m not sure if the paedomorphosis argument makes sense here. My general way of thinking about paedomorphosis is that it is a statement about adult morphology, specifically that an adult descendant has a morphology that resembles a juvenile of and ancestor (due to a shift in ontogeny that results in retardation of phenotypic development and/or size increase). However, I think you are suggesting that Exaeretodon juveniles resembled adults of Diademodon, based on your comparison of your hypothetical small Exaeretodon to a larger Diademodon. Is that correct? If so, the scenario you’re describing would seem like it would entail a shift of ontogeny towards more ‘adult’ stages (i.e., young Exaeretodon have been shifted towards a more adult morphology compared to Diademodon), which would seem to be more a case of peramorphosis. Your results seem to indicate that small Exaerertodon don’t resemble large Diademodon, so the scenario doesn’t seem to be correct. However, do small Exaeretodon resemble small Diademodon? If so, then I think you would have a good argument for a conserved developmental pattern since they start off with similar shapes early in ontogeny, and undergo similar somewhat allometric changes. To test really test this though, I think you would need to add data for a small Diademodon and a large Exaeretodon to Table 5 so you could check whether the species have similar starting shapes in ontogeny and similar ending shapes.

Line 828: Add a period after tetragonus. Also, I don’t understand your argument here either. You just said that juvenile Exaeretodon have different proportions for jaw muscle attachment areas than large Diademodon, but then you seem to suggest a similar (omnivorous) diet in small Exaeretodon and large Diademodon on the basis of similar proportions of jaw muscle attachment areas. So I don’t really see how you can argue for Exaeretodon recapitulating gomphodont dietary evolution over the course of its ontogeny with the data you have. It seems like to do that you would want to show that small Exaeretodon have skull proportions similar to larger Diademodon, implying a similar omnivorous diet, followed by Exaeretodon continuing to change skull proportions to a new shape that is presumably better suited for herbivory.

Line 845: It is unfortunate that you don’t have temporal length data for Diademodon. It seems like it would be key to testing whether the change in relative growth of the zygoma vs. the temporal region as indeed a synapomorphy of Eucynodontia. Is there any possibility of getting relevant data from previous allometric studies of Diamdemodon or other sources (like data from specimens or photos)?

---

## Round 0.4 · Minor Revisions

I have heard back from one of the reviewers and they only have a few more suggestions. Authors are free to accept them or not, but I wanted to pass them along anyway.

I believe that afterwards the paper is ready to be finally accepted. Thank you for all your resilience going through the review process, but I firmly think it improved the paper considerably.

Reviewer 3 ·

Basic reporting

General comments: This is the second time I’ve reviewed this manuscript, and as far as I can tell, the authors have done a good job of addressing the questions and suggestions I had for the previous version. Most importantly, their discussion of evolutionary patterns and heterchrony in the discussion, while more conservative, also seem more logical and consistent with the general literature on herterochrony. I don’t have many remaining suggestions for improvement, but I do have a few things that the authors might consider changing.

1) I think the section called ‘referred characters’ should be called Diagnosis or Revised Diagnosis. It clearly seems to be a diagnosis, and I haven’t seen diagnoses called ‘referred characters’ previously.

2) It’s not a critical addition, but the authors should consider including a few sentences in the discussion about the implications of their work for the ‘CREA’ allometric pattern that is common in mammals (i.e., positive allometry of the facial region). Although the pattern seems common in mammals, a few studies have suggested explicitly or implicitly that this is not uniform across non-mammalian synapsids, and that non-mammalian cynodonts don’t seem to show this pattern. The results here (i.e., negative allometry of the snout/positive allometry of the temporal region) fit the narrative of cynodonts not showing CREA, and this is quite interesting given their close relationship with mammals.

3) Because Fig. 9 is used as part of the framework for discussing dietary evolution in cynodonts, it would be helpful to include dietary information on it. I think this could be as simple as adding some leaf/meat icons next to the taxon names.

Finally, I recommend that the authors give the paper a final read to make sure all typos, etc. are taken care of. There were a couple places I noted wording issues (e.g., line 314 seems to have an unnecessary question mark; there seems to be a word missing in the discussion of the transverse process at line 601).

Experimental design

see above

Validity of the findings

see above

Additional comments

see above

---

## Round 0.5 · accepted · Accept

Thanks very much for all efforts that you have put into manuscript. I have no other choice but to highly recommend it for publication as is after this final round of revision.

Congratulations